# TPDiff: Temporal Pyramid Video Diffusion Model

**Lingmin Ran & Mike Zheng Shou**[*]
Department of Electrical and Computer Engineering
National University of Singapore
University Hall, Singapore
{andy.ran, mikeshou}@nus.edu.sg

## Abstract

The development of video diffusion models unveils a significant challenge: the substantial computational demands. To mitigate this challenge, we note that the reverse process of diffusion exhibits an inherent entropy-reducing nature. Given the inter-frame redundancy in video modality, maintaining full frame rates in high-entropy stages is unnecessary. Based on this insight, we propose TPDiff, a unified framework to enhance training and inference efficiency. By dividing diffusion into several stages, our framework progressively increases frame rate along the diffusion process with only the last stage operating on full frame rate, thereby optimizing computational efficiency. To train the multi-stage diffusion model, we introduce a dedicated training framework: stage-wise diffusion. By solving the partitioned probability flow ordinary differential equations (ODE) of diffusion under aligned data and noise, our training strategy is applicable to various diffusion forms and further enhances training efficiency. Comprehensive experimental evaluations validate the generality of our method, demonstrating 50% reduction in training cost and 1.5x improvement in inference efficiency.

## 1 Introduction

With the development of diffusion models, video generation has achieved significant breakthroughs. The most advanced video diffusion models (OpenAI, 2024; KuaiShou, 2024; Polyak et al., 2024) not only enable individuals to engage in artistic creation but also demonstrate immense potential in other fields like robotics (Songming et al., 2024) and virtual reality (Melech et al., 2023). Despite the powerful performance of video diffusion models, the complexity of jointly modeling spatial and temporal distribution makes their training costs prohibitively high (OpenSora, 2023; Zhuoyi et al., 2024; Weijie et al., 2024). Moreover, as the demand for long videos increases, the training and inference costs will continue to scale accordingly.

To alleviate this problem, researchers propose a series of approaches to increase training and inference efficiency. Show-1 (Zhang et al., 2023) and Lavie (Wang et al., 2023) adopts cascaded framework to model temporal relations at low resolution and apply super-resolution to improve the final video resolution. However, the cascaded structure leads to error accumulation and significantly increases the inference time. SimDA (Zhen et al., 2024) proposes a lightweight model which replaces Attention (Vaswani et al., 2017) with 3D convolution (Du et al., 2015) to model temporal relationship. Although convolution is computationally efficient, DiT (William & Xie, 2023) demonstrates that attention-based model is scalable and achieves better performance as the volume of data and model parameters increases. Recently, (Yang et al., 2024) introduces an interesting work: pyramid flow. This method proposes spatial pyramid: it employs low resolution during the early diffusion steps and gradually increases the resolution as the diffusion process proceeds. It avoids the need to always maintain full resolution, significantly reducing computational costs.

However, pyramid flow has several problems: 1) It only demonstrates its effectiveness under flow matching (Xingchao et al., 2022) and does not explore its applicability to other diffusion forms like

---
[*]Corresponding Author.

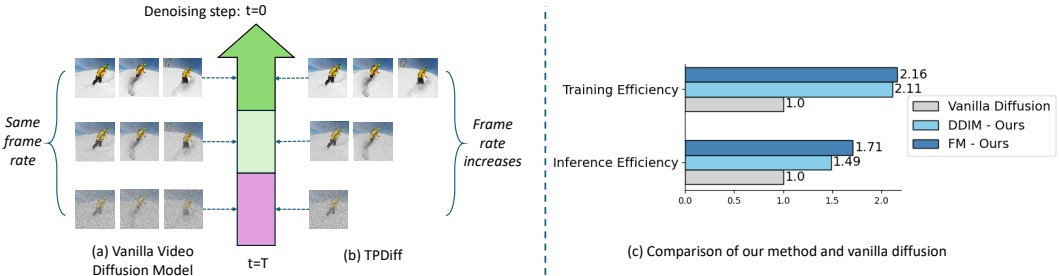

Figure 1: ***Overview of our method.*** Our method employs progressive frame rates, which utilizes full frame rate only in the final stage as shown in (a) and (b), thereby largely optimizing computational efficiency in both training and inference shown in (c).

denoising diffusion implicit models (DDIM) (Jiaming et al., 2020). 2) It formulates video generation in an auto-regressive manner which significantly reduces inference speed. 3) The feasibility of modeling temporal relationship in a pyramid-like structure remains unexplored.

To solve the problems, *we propose TPDiff, a general framework to accelerate training and inference speed.* Our method is inspired by the fact that video is a highly redundant modality (Ma et al., 2019), as consecutive frames often contain minimal variations. Additionally, in a typical diffusion process, latents in the early timesteps contain limited informative content and the temporal relations between frames are weak, which makes maintaining full frame rate throughout this process unnecessary. Based on this insight, we propose temporal pyramid: 1) In our method, the frame rate progressively increases as the diffusion process proceeds as shown in Fig. 1. Unlike previous works (Wang et al., 2023; Zhang et al., 2023) require an additional temporal interpolation network, we adopt a single model to handle different frame rates. To achieve this, we divide the diffusion process into multiple stages, with each stage operating at different frame rate. 2) To train the temporal pyramid model, we solve the partitioned probability flow ordinary differential equations (ODE) (Yang et al., 2021; Cheng et al., 2022) by leveraging data-noise alignment and reach a unified solution for various types of diffusion. 3) Our experiments show our method is generalizable to different diffusion forms, including flow matching and DDIM, accelerating training and inference compared to vanilla diffusion models as shown in Fig. 1. The core contributions of this paper are summarized as follows:

- We introduce temporal pyramid video diffusion model, a generalizable framework aiming at enhancing the efficiency of training and inference for video diffusion models. By employing progressive frame rates across different stages of the diffusion process, the framework achieves substantial reductions in computational cost.

- We design a dedicated training framework: stage-wise diffusion. We solve the decomposed probability flow ODE by aligning noise and data. The solution is applicable to different diffusion forms, enabling flexible and seamless extension to various video generation frameworks.

- Our experiments demonstrate that the proposed method can be applied across various diffusion frameworks, achieving performance improvement, 2.1x faster training and 1.5x faster inference.

## 2 RELATED WORKS

**Generative Video Models**. The field of video generation has witnessed significant progress recently due to the advancement of diffusion models (Sohl-Dickstein et al., 2015; Dhariwal & Nichol, 2021) These models generate videos from text descriptions or images. Most methods develop video models based on powerful text-to-image models like Stable Diffusion (Rombach et al., 2021), adding extra layers to capture cross-frame motion and ensure consistency. Among these, Tune-A-Video (Wu et al., 2023) employs a causal attention module and limits training module to reduce computational costs. AnimateDiff (Guo et al., 2024) utilizes a plug-and-play temporal module to enable video generation on personalized image models (AI). Recently, DiT models (Xin et al., 2024; OpenSora, 2023) pushes the boundaries of video generation. Commercial products (OpenAI, 2024; KuaiShou,

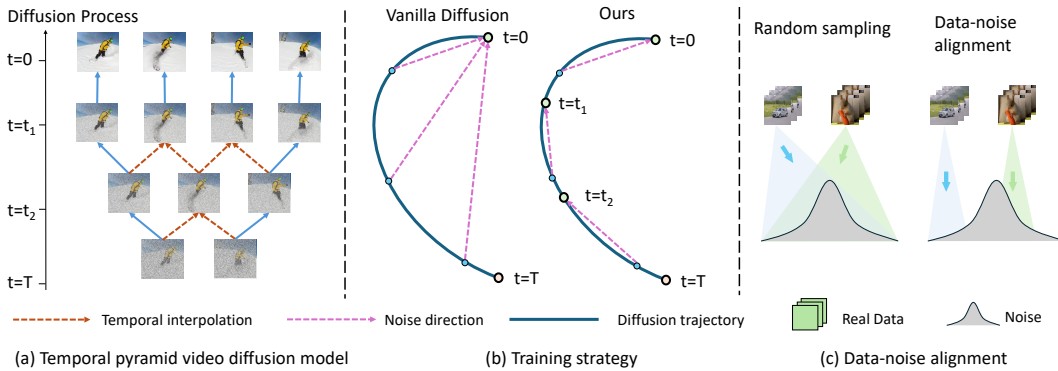

Figure 2: *Methodology*. a) ***Pipeline of temporal pyramid video diffusion model.*** We divide diffusion process into multiple stages with increasing frame rate. In each stage, new frames are initially temporally interpolated from existing frames. b) ***Our training strategy: stage-wise diffusion.*** In vanilla diffusion models, the noise direction along the ODE path points toward the real data distribution. In stage-wise diffusion, the noise direction is oriented to the end point of the current stage. c) ***Data-Noise Alignment.*** For every training sample, vanilla diffusion training randomly samples noises across the entire noise distribution, resulting in stochastic ODE path during training. In contrast, our method samples noises in the closest range, making the ODE path approximately deterministic during training

2024; Luma, 2024) and open-source works (Weijie et al., 2024; Zhuoyi et al., 2024; OpenSora, 2023) demonstrate remarkable performance by scaling up DiT pretraining. Although DiT achieves significant performance improvements, its training cost escalates to an unaffordable level, hindering the development of video generation.

**Temporal Pyramid**. The complex temporal structure of videos raises a challenge for generation and understanding. SlowFast (Christoph et al., 2019) simplifies video understanding by utilizing an input-level frame pyramid, where frames at different levels are sampled at varying rates. Each level is independently processed by a separate network, with their mid-level features interactively fused. This combination of the frame pyramid enables SlowFast to efficiently manage the variability of visual tempos. Similarly, DTPN (Da Zhang & Wang., 2018) employs different frame-per-second (FPS) sampling to construct a pyramidal representation for videos of arbitrary length. Temporal pyramid network (Yang et al., 2020) leverages the feature hierarchy to handle the variance of temporal information. It avoids to learn visual tempos inside a single network, and only need frames sampled at a single rate at the input-level. Although the effectiveness of temporal pyramid have been validated in video understanding, its application in generation remains under-explored.

# 3 METHOD

## 3.1 PRELIMINARY

**Denoising Diffusion Implicit Models**   DDIM (Jiaming et al., 2020) extends DDPMs (Ho et al., 2020) by operating in the latent space. Similar to DDPM, in the forward process, DDIM transforms real data $x_0$ into a series of intermediate sample $x_t$, and eventually the Gaussian noise $\epsilon \sim N(0, I)$ according to noise schedule $\overline{\alpha}_t$:

$$x_t = \sqrt{\overline{\alpha}_t}x_0 + \sqrt{1 - \overline{\alpha}_t}\epsilon, \epsilon \sim N(0, I), \tag{1}$$

where $t \sim [1, T]$ and $T$ denotes the total timesteps. After adding noise to the latent, we usually train a neural network $\epsilon_\theta$ to predict the added noise. Formally, $\epsilon_\theta$ is trained using following objecive:

$$\min_\theta E_{x_t, \epsilon \sim N(0, I), t \sim \text{Uniform}(1, T)} \|\epsilon - \epsilon_\theta(x_t, t)\|_2^2. \tag{2}$$

Given a pretrained diffusion model $\epsilon_\theta$, one can generate new data by solving the corresponding probability flow ODE (Yang et al., 2021). DDIM is essentially a first-order ODE solver, which

formulates a denoising process to generate $x_{t-1}$ from a sample $x_t$ via:

$$x_{t-1} = \alpha_{t-1}\left(\frac{x_t - \sqrt{1-\alpha_t}\epsilon_\theta(x_t,t)}{\alpha_t}\right) + \sqrt{1-\alpha_t}\epsilon_\theta(x_t,t),\tag{3}$$

where $\alpha_t = \frac{\overline{\alpha}_t}{\overline{\alpha}_{t-1}}$.

**Flow Matching** Flow-based generative models aim to learn a velocity field $v_\theta$ that transports Gaussian noise $\epsilon \sim N(0,I)$ to the distribution of real data $x_0$. Flow matching (Xingchao et al., 2022) adopts linear interpolation between noise $\epsilon$ and data $x_0$:

$$x_t = (1-t)x_0 + t\epsilon, \epsilon \sim N(0,I).\tag{4}$$

It trains a neural network $\epsilon_\theta$ to match the velocity field and then solves the ODE for a given boundary condition $\epsilon$ to obtain the flow. The flow matching loss function is as follows:

$$\min_\theta E_{x,\epsilon \sim N(0,I), t\sim \text{Uniform}(1,T)} \|(\epsilon - x_0) - v_\theta(x_t,t)\|_2^2.\tag{5}$$

## 3.2 Temporal pyramid diffusion

The core module of existing video diffusion models, attention (Vaswani et al., 2017), exhibits quadratic complexity with respect to sequence length. Our goal is to reduce the sequence length in video generation and decrease the computational cost. Our method is based on two key insights: 1) There is considerable redundancy between consecutive video frames. 2) the early stages of the diffusion process remain at low signal-to-noise ratio (SNR), resulting in minimal information content. It suggests that operating at full frame rate during these initial timesteps is unnecessary. Based on these insights, we propose temporal pyramid video diffusion as shown in Fig. 2 (a). Compared to traditional video diffusion model using fixed frame rate, our framework progressively increases the frame rate as the denoising proceeds.

In detail, we divide the diffusion process into multiple stages, each characterized by a distinct frame rate, and employ a single model to learn data distributions across all stages. We create $K$ stages $\{[t_k, t_{k-1}]\}_{k=K}^1$ where $0 < t_1 < t_2 ... < t_{K-1} < T$, $T$ denotes the total timesteps. The frame rate at the $k^{th}$ stage is reduced to $\frac{1}{2^{k-1}}$ of the original one. It ensures that only the last stage operates at full frame rate, thereby optimizing computational efficiency. Despite efficiency, the vanilla diffusion model does not support multi-stage training and inference. Therefore, the remaining challenges are: 1) How to train the multi-stage diffsion model in a unified way, which will be introduced in Section 3.3 and Section 3.4, 2) How to perform inference, which will be discussed in Section 3.6.

## 3.3 Training strategy

In stage $k$, we denote $(s_k, e_k)$ as the start and end timestep, $\hat{x}_{s_k}$ and $\hat{x}_{e_k}$ as start and end point. The objective of training is to transport distribution of $\hat{x}_{s_k}$ to $\hat{x}_{e_k}$ at every stage. To achieve the objective, the key is to obtain stage-wise 1) target, *i.e.* $\epsilon$ in DDIM and $\frac{dx_t}{dt}$ in flow matching, and 2) intermediate latents $x_t$ where $t \in [s_k, e_k]$ (Fu-Yun et al., 2024; Hanshu et al., 2024). In the following, we will introduce a unified training framework named stage-wise diffusion.

**Stage-wise Diffusion** To ensure generality, recognizing different diffusion frameworks share a similar formulation as shown in Equation 1 and Equation 4, we present a unified diffusion form:

$$x_t = \gamma_t x_0 + \sigma_t \epsilon,\tag{6}$$

where the form of $\gamma_t$ and $\sigma_t$ depend on diffusion framework selected. Our derivation is based on Equation 6, without constraining the parameterization of $\gamma_t$ and $\sigma_t$. Considering continuity between stages with distinct frame rates, we obtain $\hat{x}_{s_k}$ and $\hat{x}_{e_k}$ by:

$$\hat{x}_{s_k} = \gamma_{s_k} Up(Down(x_0, 2^{k+1}), 2) + \sigma_{s_k}\epsilon,\tag{7}$$

$$\hat{x}_{e_k} = \gamma_{e_k} Down(x_0, 2^k) + \sigma_{e_k}\epsilon,\tag{8}$$

where $\epsilon \sim N(0,I)$, $Down(\cdot, 2^k)$ and $Up(\cdot, 2^k)$ are downsampling and upsampling $2^k$ times along temporal axis. We derive the start point of current stage from the end point of preceding stage

in Equation 7 to bridge adjacent stages, which is crucial for inference and will be introduced in Section 3.6. However, this design also leads to boundary distribution shift and we cannot directly obtain training target from Equation 7 and Equation 8. Instead, we should compute added noise in every stage with boundary condition $\hat{x}_{s_k}$ and $\hat{x}_{e_k}$. Fortunately, DPM-Solver (Cheng et al., 2022) derives the relationship between any two points, $x_s$ and $x_e$ on diffusion ODE path and this relationship can also be applied to any stage in our method. Accordingly, in stage $k$, by replacing $x_s$ with $\hat{x}_{s_k}$ and $x_e$ with $x_t$, we can express intermediate latent $x_t$ as a function of $\hat{x}_{s_k}$:

$$x_t = \frac{\gamma_t}{\gamma_{s_k}}\hat{x}_{s_k} - \gamma_t \int_{\lambda_{s_k}}^{\lambda_t} e^{-\lambda}\epsilon(x_{t_\lambda}, t_\lambda)d\lambda, \tag{9}$$

where $e_k < t < s_k$, $\lambda_t = ln\frac{\gamma_t}{\sigma_t}$, and $t_\lambda$ is the inverse function of $\lambda_t$. Equation 9 consists of two components: a deterministic scaling factor, given by $\frac{\gamma_t}{\gamma_{s_k}}$, and the exponentially weighted integral of the noise $\epsilon(x_{t_\lambda}, t_\lambda)$. If $\epsilon(x_{t_\lambda}, t_\lambda)$ is a constant in stage $k$, denoted as $\epsilon_k$, the above integral is equivalent to:

$$\begin{aligned} x_t &= \frac{\gamma_t}{\gamma_{s_k}}\hat{x}_{s_k} - \gamma_t\epsilon_k \int_{\lambda_{s_k}}^{\lambda_t} e^{-\lambda}d\lambda \\ &= \frac{\gamma_t}{\gamma_{s_k}}\hat{x}_{s_k} + \gamma_t\epsilon_k \left(\frac{\sigma_t}{\gamma_t} - \frac{\sigma_{s_k}}{\gamma_{s_k}}\right). \end{aligned} \tag{10}$$

While enforcing a constant value for $\epsilon_k$ at any stage is challenging, we can leverage data-noise alignment (Li et al., 2024) to constrain its value within a narrow range. In detail, before adding noise to video, we pre-determine the target noise distribution for each video by minimizing the aggregate distance between video-noise pairs as shown in Fig. 2 (c), thereby ensuring data-noise alignment and Equation 9 are approximately equivalent to Equation 10. The alignment process can be implemented using Scipy (Pauli et al., 2020) in one line of code as shown in Algorithm 1.

---

**Algorithm 1** Data-Noise Alignment

**Require:** Video batch $x$, random noise $\epsilon$
 1: assign_mat $\leftarrow$ `scipy.optimize.linear_sum_assignment(dist(x, ε))`
 2: $\epsilon' \leftarrow \epsilon$[assign_mat]
**Output:** $\epsilon'$

---

Our experiments in Section 4 demonstrate that this approximation is valid and does not compromise the model's performance. Notably, the computational cost introduced by alignment is negligible, and we provide the extra time required for alignment in Appendix A.2.

Through data-noise alignment, we can apply Equation 10 to any point in the stage, including the end point $\hat{x}_{e_k}$. By substituting $t = e_k$ and $x_t = \hat{x}_{e_k}$ into Equation 10, through simple transformation, we arrive at the expression for noise $\epsilon_k$ of stage $k$:

$$\epsilon_k = \frac{\frac{\hat{x}_{e_k}}{\gamma_{e_k}} - \frac{\hat{x}_{s_k}}{\gamma_{s_k}}}{\frac{\sigma_{e_k}}{\gamma_{e_k}} - \frac{\sigma_{s_k}}{\gamma_{s_k}}}. \tag{11}$$

Then we can easily get any intermediate point $x_t$ in stage $k$ by substituting $\epsilon_k$ into Equation 10. Consequently, we can compute the corresponding loss using $x_t$ and $\epsilon_k$ obtained in our method and optimize model parameters in the same way as vanilla diffusion training. Note that the above derivation does not constrain the expressions of $\gamma_t$ and $\sigma_t$, making our method applicable to different diffusion frameworks. We also note that the direction of $\epsilon_k$ points towards the end point of the current stage rather than the final target in vanilla diffusion models as shown in Fig. 2 (b). By reducing the distance between intermediate points and their target points, our method facilitates the training process and further accelerates model convergence.

## 3.4 PRACTICAL IMPLEMENTATION

In practice, for diffusion framework whose ODE path is curved, like DDIM, we can substitute $\gamma_t = \sqrt{\bar{\alpha}_t}$ and $\sigma_t = \sqrt{1 - \bar{\alpha}_t}$ into Equation 10 and Equation 11 to obtain $x_t$ and $\epsilon_k$. For flow

matching, unlike DDIM which requires the prior distribution to be standard Gaussian distribution, it can transport any prior distribution to other distributions. Thus, we can model each stage as a complete flow matching process (Yang et al., 2024), resulting in a simpler expression:

$$x_t = (1 - t')\hat{x}_{e_k} + t'\hat{x}_{s_k}, \tag{12}$$

where $t' = \frac{t - e_k}{s_k - e_k}$. And The objective of stage $k$ is:

$$\frac{dx_t}{dt'} = \hat{x}_{s_k} - \hat{x}_{e_k} \tag{13}$$

One aspect pyramid flow overlooks is the noise-data aligment, leading to increased variance in the prior distribution, thereby hindering model convergence. Notably, if we model each stage as a complete DDIM process, the model fails to converge. This is because it is exceedingly challenging for a single model to fit multiple curved ODE trajectories.

In conclusion, we visualize the training process of our method in Algorithm 2.

---

**Algorithm 2** Stage-wise Diffusion

---

**Require:** Training dataset $D$, Number of stages $K$, Diffusion type DDIM or Flow Matching, Model $\epsilon_\theta$ or $v_\theta$, Create $K$ stages $\{[s_k, e_k]\}_{k=1}^K$
 1: **repeat**
 2:  Sample $x_0 \sim D$;
 3:  Sample stage $k$ from $\{1, ...K\}$, then sample timestep $t \in [s_k, e_k)$
 4:  Sample noise $\epsilon' \in N(0, I)$ aligned with $x_0$
 5:  Add $\epsilon'$ to $x_0$ and get $\hat{x}_{e_k}$ and $\hat{x}_{s_k}$
 6:  **if** Flow Matching **then**
 7:   $x_t = (1 - t')\hat{x}_{e_k} + t'\hat{x}_{s_k}$ where $t' = \frac{t - e_k}{s_k - e_k}$
 8:   $v_k = \hat{x}_{s_k} - \hat{x}_{e_k}$
 9:   Compute loss: $\ell = ||v_\theta(x_t) - v||^2$
10:  **else**
11:   $\epsilon_k = \frac{\frac{\hat{x}_{e_k}}{\alpha_{e_k}} - \frac{\hat{x}_{s_k}}{\alpha_{s_k}}}{\frac{\sigma_{e_k}}{\alpha_{e_k}} - \frac{\sigma_{s_k}}{\alpha_{s_k}}}$
12:   $x_t = \frac{\alpha_t}{\alpha_{s_k}}\hat{x}_{s_k} + \alpha_t\epsilon_k\left(\frac{\sigma_t}{\alpha_t} - \frac{\sigma_{s_k}}{\alpha_{s_k}}\right)$
13:   Compute loss: $\ell = ||\epsilon_\theta(x_t) - \epsilon_k||^2$
14:  **end if**
15:  Update $\theta$ with gradient-based optimizer using $\nabla_\theta \ell$
16: **until** Convergence

---

### 3.5 Stage-consistent Positional Encoding

We employ RoPE (Su J, 2024) as our positional encoding. A direct application of RoPE at each stage would assign new positional indices after upsampling, causing misalignment between stages. In practice, this misalignment restricts the model to generate only small-amplitude motions.

To overcome this limitation, we propose stage-consistent positional encoding, where each frame index is multiplied by a stage-dependent factor to ensure that the same frame across different stages is assigned an identical positional encoding. Note that the first frame in every stage is shared so we do not scale it. Formally, let the diffusion process be divided into $m$ stages. For the $n$-th frame in the $i$-th stage, the positional encoding is defined as:

$$PE_i(n) = \begin{cases} \text{RoPE}\big(n \cdot 2^{(m-i)}\big), & \text{if } n > 1 \\ \text{RoPE}(n), & \text{otherwise} \end{cases} \tag{14}$$

where $\text{RoPE}(\cdot)$ denotes the rotary positional encoding function. By maintaining consistent positional encoding across stages, our strategy enables the model to capture both large-amplitude motions, significantly enhancing temporal dynamic degree.

### 3.6 Inference Strategy

After training, we can use standard sampling algorithm (Cheng et al., 2022) to solve the reverse ODE in every stage. However, we need to carefully handle the stage continuity. Upon completion

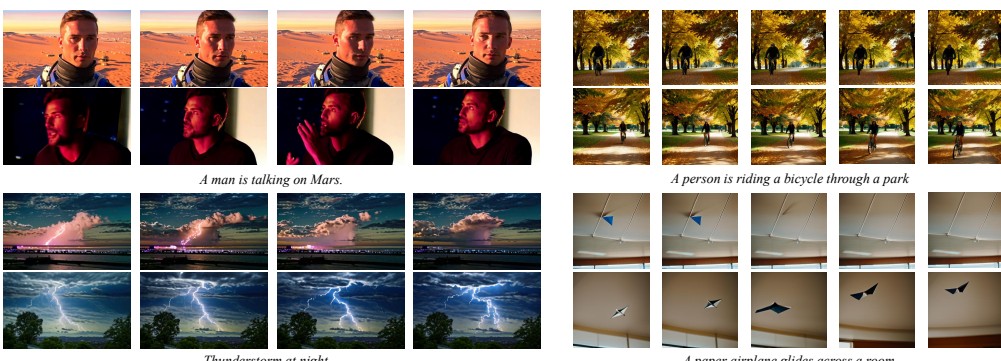

Figure 3: *Qualitative comparison*. In each pair of videos, the first row presents the results of models trained using vanilla diffusion and the second row shows the results of our method. Video pairs in the left are generated by MiniFlux-vid-ours and the remaining are generated by Animatediff-ours.

of a stage, we first upsample $e_k$ in temporal dimension to double its frame rate via interpolation. Subsequently, we scale $Up(\hat{x}_{e_k})$ and inject additional random noise to match the distribution of $\hat{x}_{s_{k-1}}$ during training. Considering the simplest scenario using nearest temporal upsampling and lowering the effect of noise, we can get $\hat{x}_{s_{k-1}}$ using the equation below (see Appendix A.1 for detailed derivations):

$$\hat{x}_{s_{k-1}} = \frac{\sqrt{2}\gamma_{s_k}}{\sigma_{s_k} + \sqrt{2}\gamma_{s_k}} Up(\hat{x}_{e_k}) + \frac{\sqrt{2}\sigma_t}{2} n', n' \sim N(0, \begin{bmatrix} 1 & -1 \\ -1 & 1 \end{bmatrix}). \tag{15}$$

## 4 EXPERIMENTS

### 4.1 EXPERIMENTAL SETTING

We implement our method in both DDIM and flow matching. Since most video diffusion models are bulit upon pretrained image models, our experiments are based on two image models: MiniFlux (Yang et al., 2024) and SD1.5 (Rombach et al., 2021). These two models are trained under flow matching and DDIM, respectively. We extend MiniFlux to MiniFlux-vid by finetuning all its parameters on video data and we follow AnimateDiff (Guo et al., 2024) to extend SD1.5 to video model. Moreover, to demonstrate the generalization ability of our approach, we finetune Wan (Wan et al., 2025) using our method. Specifically, we apply LoRA (Hu et al., 2021) to the pretrained Wan and perform training with our temporal pyramid method. The number of stages is set to 3 and each stage is uniformly partitioned in all experiments. Our experiments are conducted on NVIDIA H100 GPU. The detailed training setting is in Appendix A.4.

**Dataset** We adopt OpenVID1M (Kepan et al., 2024) as our dataset. The training video resolutions for MiniFlux-vid, AnimateDiff and Wan are 640x384, 512×512 and 832x480 respectively.

**Baselines** We compare our method with video diffusion models of the same network architecture but trained in the vanilla diffusion framework (i.e., pretrained Animatediff and MiniFlux-vid) to demonstrate that our approach does not lead to performance degradation. Note that MiniFlux-vid is trained under the flow matching framework without applying the temporal pyramid, using the same initialization and hyperparameters as in our method. To evaluate the effectiveness of our approach compared to existing methods, we compare our method with ModelScope (Jiuniu et al., 2023), OpenSora v1.3 (OpenSora, 2023), Videocrafter2 (Chen et al., 2023) and CogVideoX-5B (Zhuoyi et al., 2024). Furthermore, to compare with other efficiency-enhancing approaches, we also compare our method with SimDA (Zhen et al., 2024).

**Evaluation** We evaluate our model from two perspectives: generation quality and efficiency. To evaluate the generation quality, we adopt quantitative metrics from VBench (Huang et al., 2024) to compare our method's performance with existing models. For efficiency, we visualize the convergence curve to intuitively demonstrate training efficiency. In detail, to evaluate the convergence speed, we follow common practice (Jiuniu et al., 2023) to use validation videos from MSRVTT (Jun et al.,

| Models | #Params. | Total | QS | SC | TF | MS | DD | AQ | HA | SR | TS | OC |
|---|---|---|---|---|---|---|---|---|---|---|---|---|
| ModelScope (Jiuniu et al., 2023) | 1.7B | 75.75 | 78.05 | 89.87 | 98.28 | 95.79 | **66.39** | 52.06 | 92.40 | 43.95 | 25.37 | 26.18 |
| OpenSora 1.3 (OpenSora, 2023) | 1.1B | 75.98 | 80.14 | **97.27** | 99.12 | 99.17 | 35.28 | 60.42 | 58.20 | 49.64 | 22.47 | 20.39 |
| CogVideoX (Zhuoyi et al., 2024) | 5B | 80.91 | 82.75 | 96.78 | 98.89 | 97.73 | 59.86 | 61.98 | **98.00** | 69.90 | 25.38 | 26.66 |
| Videocrafter2 (Chen et al., 2024) | 1.5B | 80.44 | 82.20 | 96.76 | **99.77** | 99.51 | 37.22 | 63.13 | 88.00 | 65.65 | 25.84 | 25.47 |
| SimDA (Zhen et al., 2024) | 1B | 76.62 | 81.63 | 94.21 | 98.78 | 98.16 | 32.63 | 49.12 | 76.00 | 46.72 | 24.95 | 25.42 |
| RepVideo (Si et al., 2025) | 4B | 81.94 | 82.70 | 78.91 | 99.16 | 98.13 | 57.78 | 62.40 | 98.00 | 74.74 | 25.31 | 26.96 |
| Mochi-1 (Team, 2024) | 10B | 80.13 | 82.64 | 70.08 | 99.40 | 99.02 | 61.85 | 56.94 | 94.60 | 69.24 | 23.65 | 25.15 |
| Step-Video-T2V (Ma et al., 2025) | 30B | 81.83 | **84.46** | 71.28 | 99.40 | 99.08 | 53.06 | 61.23 | 94.00 | 71.47 | 26.01 | 27.12 |
| MAGI-T2V-4.5B (ai et al., 2025) | 4.5B | 77.92 | 80.98 | 65.68 | 99.71 | 99.48 | 20.28 | 62.42 | 84.20 | 57.76 | 23.03 | 25.56 |
| Open-Sora-2.0 (Peng et al., 2025) | 11B | 81.71 | 82.10 | 80.14 | 99.40 | 99.49 | 20.74 | 64.33 | 95.40 | 76.18 | 25.91 | 27.57 |
| AnimateDiff (Guo et al., 2024) | 1.8B | 80.27 | 82.90 | 95.30 | 98.75 | 97.76 | 54.83 | 67.16 | 92.60 | 34.60 | 26.03 | 27.04 |
| AnimateDiff - Ours | 1.8B | 80.76 | 81.80 | 96.87 | 98.49 | 98.12 | 58.71 | 62.19 | 94.20 | 44.18 | 25.14 | 27.36 |
| MiniFlux-vid - Vanilla | 9B | 81.54 | 82.14 | 97.12 | 99.34 | 98.67 | 61.29 | 60.34 | 96.60 | 67.49 | 25.32 | 27.42 |
| MiniFlux-vid - Ours | 9B | **81.95** | 82.81 | 96.94 | 99.64 | 98.94 | 64.37 | 61.92 | 98.00 | 65.81 | 25.94 | **27.94** |
| Wan (Wan et al., 2025) | 1.3B | 84.26 | 84.28 | 96.34 | 99.49 | 97.44 | 37.76 | 76.46 | 95.20 | 76.66 | 24.17 | 27.07 |
| Wan - Ours | 1.3B | 84.33 | 83.74 | 97.86 | 99.07 | 92.19 | 42.13 | 73.91 | 95.4 | 74.47 | 25.86 | 27.19 |

Table 1: Comparison of video generation quality across baselines and our method. #Params. indicates the number of total parameters. QS: Quality Score, SC: Subject Consistency, TF: Temporal Flickering, MS: Motion Smoothness, DD: Dynamic Degree, AQ: Aesthetic Quality, HA: Human Action, SR: Spatial Relationship, TS: Temporal Style, OC: Overall Consistency. Because Wan-Ours is finetuned from a pretrained Wan model using LoRA, rather than being trained from scratch, it is therefore excluded from comparisons with other models.

2016) for zero-shot generation evaluation. We systematically compute the FVD (Thomas et al., 2018) value during training and present the FVD-GPU hours curve to demonstrate the training efficiency of our method. We also report the average inference time to validate the inference efficiency.

## 4.2 QUANTITATIVE RESULTS

| #Stages | Partition | Red. rate | w. Align | TS (↑) | Train Speedup | Infer Speedup |
|---|---|---|---|---|---|---|
| 3 | 1-1-1 | 2 | Yes | **80.76** | **2.16×** | 1.49× |
| 3 | 1-1-1 | 2 | No | 79.16 | 1.75× | 1.49× |
| 4 | 1-1-1-1 | 2 | Yes | 80.14 | 1.82× | 1.65× |
| 5 | 1-1-1-1-1 | 2 | Yes | 80.03 | 1.71× | 1.74× |
| 3 | 2-1-1 | 2 | Yes | 79.82 | 1.92× | 1.62× |
| 3 | 1-1-2 | 2 | Yes | **80.94** | 1.62× | 1.36× |
| 3 | 1-1-1 | 4 | Yes | 80.12 | 1.38× | **1.79×** |
| 3 | 1-1-1 | 4-2 | Yes | 80.65 | 2.01× | 1.54× |
| 3 | 1-1-1 | 2-4 | Yes | 80.18 | 1.76× | **1.68×** |

Table 2: Ablation study on temporal pyramid design.

| Model | Method | Latency(s)↓ |
|---|---|---|
| MiniFlux-vid | Vanilla | 20.79 |
| | **Ours** | **12.18** (1.71×) |
| AnimateDiff | Vanilla | 6.01 |
| | **Ours** | **4.04** (1.49×) |
| Wan | Vanilla | 50.52 |
| | **Ours** | **27.76** (1.82×) |

Table 3: Inference efficiency comparison. Timestep = 30.

Tab. 1 shows quantitative comparison between our method and baselines. Compared to existing method, our approach achieves improvements in most aspects, particularly in temporal-related metrics such as overall consistency (OC) and dynamic degree (DD). This indicates that vanilla video diffusion models contain substantial temporal redundancy, whereas our method effectively eliminates these redundancies and thereby exhibits stronger temporal modeling capability. Figure. 4 (a) and (b) shows that our method achieves speedup of 2x and 2.13x in training compared to vanilla diffusion models. This acceleration primarily stems from two factors:

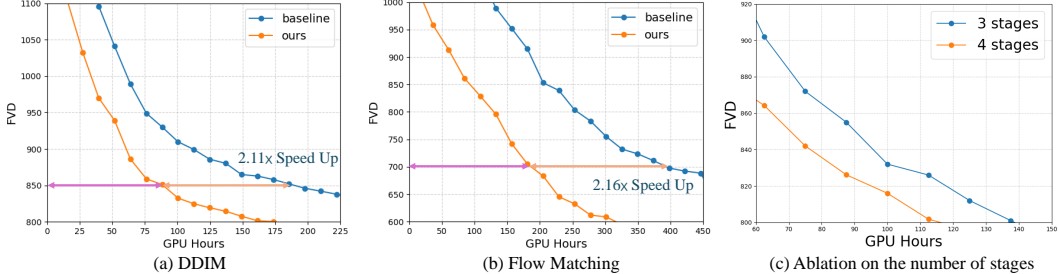

Figure 4: (a) & (b) *Convergence curve of vanilla diffusion and our method on (a) DDIM, (b) Flow Matching*. We illustrate the FVD of two methods with different GPU hours consumed. Our method achieves higher training efficiency compared to vanilla approachs. (c) *Ablation study of stages.* Increasing the number of stages leads to improved efficiency.

1) Noise-data alignment: By aligning noise with data, we reduce the randomness in training. The model learns a nearly deterministic ODE path rather than the expectation of multiple intersecting ODE paths.

2) Reduced average sequence length. Since the computational complexity of attention mechanism scales quadratically with sequence length, our method requires significantly less computational complexity on average. For example, to generate a video of length $T$, the averaged computational cost of attention modules in our method is halved, reducing to $\frac{1}{3}(T^2 + (\frac{T}{2})^2 + (\frac{T}{4})^2) \approx 0.44T^2$ compared to $T^2$ in vanilla diffusion model. This advantage is also reflected in the faster inference speed as shown in Tab. 3.

### 4.3 QUALITATIVE RESULT

As shown in Fig. 3, we show qualitative comparison between our method and vanilla video diffusion models. The results generated by our method are presented in the second column and the outputs of our baseline are displayed in the first column. Evidently, our approach is able to generate videos with better semantic accuracy and larger motion. For instance, under prompt "A man is talking on Mars", the baseline generates a person merely shaking their head without speaking, failing to fully adhere to the prompt. In contrast, our approach accurately generates the specified actions, demonstrating superior alignment with the given prompt. Moreover, for AnimateDiff, the baseline generates videos that are nearly static, whereas our approach achieves motion with a more natural and reasonable amplitude. You can find more qualitative results in the supplementary material and Appendix A.3.

### 4.4 ABLATION STUDY

We conduct ablation study to assess the contribution of different design choices in our method. Tab. 2 summarizes the results, measured in terms of the total score (TS) from VBench (Huang et al., 2024).

**Frame-rate reduction schedule.** Beyond using a fixed reduction rate of 2 across all stages, we experiment with non-uniform reduction schedules such as 4-2 and 2-4. Here, x–y denotes a non-uniform reduction schedule where the frame rate is reduced by a factor of x from Stage 1→2 and by a factor of y from Stage 2→3. A more aggressive reduction at early stages (4-2) yields competitive total score (80.65) and achieves the second-highest training speedup (2.01×), suggesting that aggressively shrinking the sequence length early can reduce compute without severely harming the final quality. Conversely, a more aggressive reduction at the last stage (2-4) keeps quality relatively stable but trades part of the inference efficiency (1.68×) against slower refinement. Overall, while non-uniform schedules offer flexibility, the uniform schedule remains the most robust across both quality and compute.

**Trade-off between training and inference efficiency.** A notable trend is that configurations offering the highest training acceleration do not always provide the highest inference acceleration. For example, the 4-stage and 5-stage variants achieve higher inference speedups (1.65× and 1.74×) due to

shorter effective sequence lengths in later stages but have slower training convergence compared to the default 3-stage setup. Similarly, using a higher reduction rate of 4 achieves the best inference speedup (1.79×) but at the cost of lower training efficiency (1.38×) and mildly degraded quality (80.12). These results again highlight that overly aggressive downsampling helps inference but slows training and harms temporal fidelity.

**Importance of late-stage refinement.** Configurations allocating more steps to the last stage (1-1-2) yield the highest TS score (80.94), supporting our observation that high–frame-rate refinement is crucial. In contrast, shifting more steps to the earliest stage (2-1-1) harms quality, likely because coarse low-frame-rate representations alone cannot capture sufficient temporal dynamics.

**Effectiveness of data–noise alignment.** Removing our proposed alignment mechanism consistently reduces performance ($80.76 \rightarrow 79.16$), confirming its critical role in stabilizing the hierarchical denoising trajectory across variable frame rates.

### 4.5 Zero-shot long video generation

Although our model is trained on fixed-length video clips, we observe that it can generate longer videos in a zero-shot manner, without any retraining or architectural modification. We attribute this capability to the design of our temporal pyramid: during training, the model naturally encounters multi–frame-rate trajectories, since each stage operates at a different temporal resolution. This exposure implicitly equips the model with robustness to varying temporal lengths and enables extrapolation to longer sequences at inference time. We provide qualitative examples as well as quantitative evaluation on longer video generation in Sec. A.6.

## 5 Conclusion

In this paper, we propose a general acceleration framework for video diffusion models. We introduce TPDiff, a framework that progressively increases the frame rate along the diffusion process. Moreover, we design a dedicated training framework named stage-wise diffusion, which is applicable to any form of diffusion. Our experiments demonstrate that our method accelerates both training and inference on different frameworks.

## 6 Acknowledge

This research is supported by the Ministry of Education, Singapore, under its Academic Research Fund Tier 2 (Award No: MOE-T2EP20124-0012).

### Ethics Statement

This work adheres to the Code of Ethics. Our study does not involve human subjects, sensitive personal data, or applications with foreseeable harmful consequences. All datasets used are publicly available and described in detail in Section 4, with careful consideration of potential biases and fairness concerns. The proposed methods are intended solely for academic research, and we have taken steps to ensure compliance with ethical research standards and legal requirements.

### Reproducibility Statement

We have taken several steps to ensure the reproducibility of our work. Section 4 provides detailed descriptions of the model architecture, training objectives, and evaluation protocols. Additional implementation details, hyperparameter settings are included in the Appendix A.4, while the supplementary materials contain further results.

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

# A    APPENDIX

## A.1    DERIVATION

This section provides derivation for Equation. 22. Our derivation primarily follows pyramid flow (Yang et al., 2024), and we extend it to the temporal dimension. According to Equation. 8 and Equation. 7:

$$\hat{x}_{s_k} \sim N(\gamma_{s_k} Up(Down(x_0, 2^{k+1})), \sigma_{s_k}^2 I)$$
$$Up(\hat{x}_{e_{k+1}}) \sim N(\gamma_{e_{k+1}} Up(Down(x_0, 2^{k+1})), \sigma_{e_{k+1}} I) \tag{16}$$

Spatial pyramid has demonstrate that stages can be smoothly connected by renoising the endpoint of the last stage. Renoising process can be expressed as:

$$\hat{x}_{s_k} = \frac{\gamma_{s_k}}{\gamma_{e_{k+1}}} Up(\hat{x}_{e_{k+1}}) + \alpha n', n \sim N(0, \Sigma') \tag{17}$$

where the rescaling coefficient $\frac{s_k}{e_{k+1}}$ allows the means of these distributions to be matched, and $\alpha$ is the noise weight. Additionally, we need to match the covariance matrices:

$$\frac{\gamma_{s_k}^2}{\gamma_{e_{k+1}}^2} \sigma_{k+1}^2 \Sigma + \alpha^2 \Sigma' = \sigma_{s_k}^2 I. \tag{18}$$

we consider the simplest interpolation: nearest neighbor temporal upsampling. Then we can get upsampling $\Sigma$ and noise's covariance matrix $\Sigma'$ has the same structure as $\Sigma$:

$$\Sigma_{block} = \begin{pmatrix} 1 & 1 \\ 1 & 1 \end{pmatrix} \implies \Sigma' = \begin{pmatrix} 1 & \gamma \\ \gamma & 1 \end{pmatrix} \tag{19}$$

To ensure $\Sigma'$ is semidefinite, $\gamma \in [-1, 0]$. Then we solve Equation. 18 and Equation. 19 by considering the equality of their diagonal and non-diagonal elements and get the solution:

$$\gamma_{e_{k+1}} = \frac{\gamma_{s_k} \sqrt{1-\gamma}}{\sigma_{s_k} \sqrt{-\gamma} + \gamma_{s_k} \sqrt{1-\gamma}}, \quad \alpha = \frac{\sigma_{s_k}}{\sqrt{1-\gamma}} \tag{20}$$

To reduce the affect of noise, let $\gamma = -1$ and substitute it into Equation. 20, we can get:

$$\gamma_{e_{k+1}} = \frac{\sqrt{2}\gamma_{s_k}}{\sigma_{s_k} + \sqrt{2}\gamma_{s_k}}, \alpha = \frac{\sqrt{2}\sigma_t}{2} \tag{21}$$

We can finally obtain Equation. 22:

$$\hat{x}_{s_{k-1}} = \frac{\sqrt{2}\gamma_{s_k}}{\sigma_{s_k} + \sqrt{2}\gamma_{s_k}} Up(\hat{x}_{e_k}) + \frac{\sqrt{2}\sigma_t}{2} n' \tag{22}$$

where:

$$n' \sim N(0, \begin{bmatrix} 1 & -1 \\ -1 & 1 \end{bmatrix}) \tag{23}$$

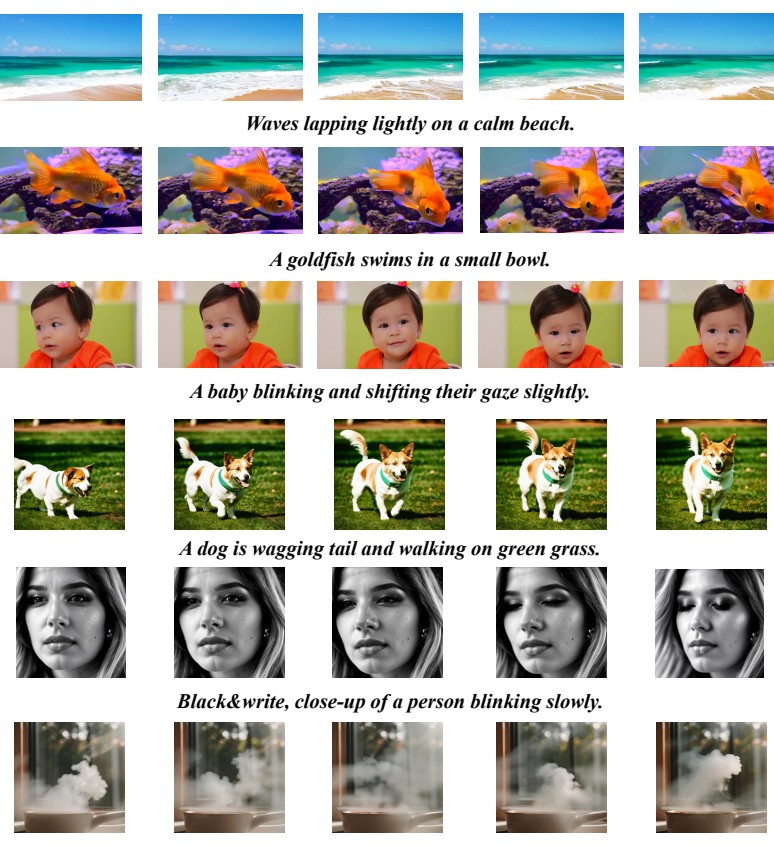

*Waves lapping lightly on a calm beach.*

*A goldfish swims in a small bowl.*

*A baby blinking and shifting their gaze slightly.*

*A dog is wagging tail and walking on green grass.*

*Black&write, close-up of a person blinking slowly.*

*Steam rises from a hot cup of coffee.*

Figure 1: ***Qualitative results***. Results in lines 1–3 are generated by MiniFlux-vid and in lines 4–6 are generated by AnimateDiff. You can find more qualitative videos in our supplementary material.

| Batch Size | 4 | 8 | 16 |
|---|---|---|---|
| Latency (ms) | 5.1 | 9.6 | 17.8 |

Table 1: Computational overhead of data-noise alignment on different batch size.

## A.2 COMPUTATIONAL OVERHEAD OF DATA-NOISE ALIGNMENT

We evaluate the runtime overhead of data-noise alignment under different batch sizes. Notably, the alignment is performed on CPUs, incurring no additional GPU memory consumption. The evaluation results is shown in Tab. 1.

## A.3 QUALITATIVE RESULTS

We show more qualitative results in Fig. 1. You can find more qualitative videos in our supplementary material.

## A.4 TRAINING SETTING

For Miniflux-vid and Animatediff training, we utilize the AdamW optimizer with a learning rate of $1e^{-4}$ and a batch size of 128. Two models are trained for 50k steps using 8 H100 GPUs. For Wan finetuning, we set LoRA rank to 128, using AdamW optimizer with a learning rate of $5e^{-5}$ to finetune for 10k steps using 8 H100 GPUS.

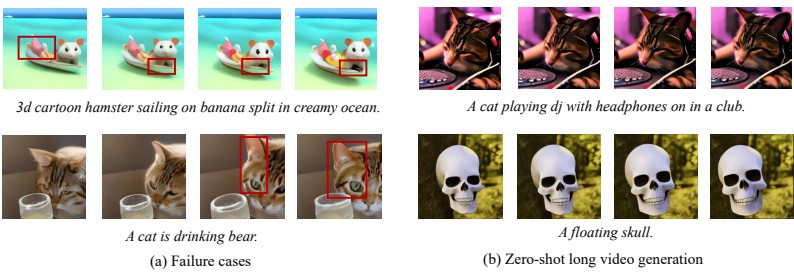

*3d cartoon hamster sailing on banana split in creamy ocean.*    *A cat playing dj with headphones on in a club.*

*A cat is drinking bear.*    *A floating skull.*

(a) Failure cases    (b) Zero-shot long video generation

Figure 2: ***Failure cases and zero-shot long video generation result.***

## A.5 FAILURE CASE STUDY

We present representative failure cases of our method as shown in Fig. 2 (a). In the first row, the generated hamster does not fully adhere to realistic anatomy, with noticeable artifacts around the paws. In the second row, the cat's head undergoes an abrupt change in size and appearance after turning, indicating instability in identity preservation across viewpoints. Importantly, these issues are not unique to our method. Similar artifacts such as local geometry distortions, identity drift, or physics-inconsistent motion are widely observed in existing video diffusion models.

## A.6 ZERO-SHOT LONG VIDEO GENERATION

Although our Animetediff model is trained on 16-frame clips and does not incorporate any explicit mechanism for temporal extrapolation, we find that it naturally supports zero-shot long video generation. At inference time, the same model can produce coherent 32-frame video clips without any retraining. We attribute this emergent capability to the structure of our temporal pyramid: each stage operates at progressively higher frame rates, effectively exposing the model to variable temporal resolutions during training. As a result, the model learns frame-to-frame dynamics in a way that is not tied to a fixed sequence length, making it robust to extended temporal horizons at test time. We provide qualitative long video generation result on Fig. 2 (b).

## A.7 LIMITATIONS

In this paper, we identify the temporal redundancy in video generation and propose a pyramid architecture to mitigate it. However, prior works (Christoph et al., 2019; Yang et al., 2024) point out that redundancy exists in both temporal and spatial dimensions. This work does not explore how to jointly eliminate temporal and spatial redundancy while preserving generation quality, which we leave it as future work.

## A.8 USE OF LLMs

In our workflow, large language models (LLMs) are employed solely for the purpose of polishing the writing of our manuscripts. Their role is restricted to improving grammar, clarity, and readability, without influencing the scientific content or experimental results. We emphasize that all conceptual ideas, methodological designs, and analyses are entirely developed by the authors. Thus, the LLM serves only as a language refinement tool rather than a contributor to the research itself.

