# OpenReview forum: "TPDiff: Temporal Pyramid Video Diffusion Model"
_ICLR.cc/2026/Conference — ICLR 2026 Poster_

### Official Review · Reviewer_uQWS · 2025-10-21

**Soundness:** 2
**Presentation:** 3
**Contribution:** 2
**Rating:** 4
**Confidence:** 5

**Summary:**

This paper proposes a staged frame-rate reduction strategy to accelerate the training of diffusion-based generative models. The method progressively lowers the temporal resolution during training, aiming to reduce computational cost while preserving generation quality.

**Strengths:**

S1. The core idea is intuitive and conceptually easy to grasp.
S2. The empirical results, based on three baselines, demonstrate consistent improvements and validate the effectiveness of the proposed approach.

**Weaknesses:**

W1. Lack of comprehensive baselines. The experimental comparisons are rather limited. The field of video diffusion acceleration has seen rapid development since 2023, with numerous relevant approaches proposed recently. However, this paper only compares its method against a vanilla diffusion baseline, without evaluating against other contemporary acceleration techniques. As a result, it is difficult to assess the relative advantage or practical relevance of this approach. Readers cannot determine whether the proposed strategy offers clear benefits over more recent methods.

W2. Outdated experimental setup. Most of the compared methods date back to 2024 or earlier, while the video generation domain evolves extremely fast. Many 2023-era methods are no longer representative of the current state of the art. For instance, the authors compare against Open-Sora 1.3 (2023), but Open-Sora 2.0 was released in March 2025—roughly six months before the submission deadline. This omission raises concerns about the timeliness and completeness of the evaluation. Incorporating more recent baselines would provide a fairer and more convincing assessment of the proposed contribution.

**Questions:**

Q1. Could the authors include comparisons with more recent (2024–2025) approaches in video generation acceleration to contextualize their results?

Q2. It would be beneficial to add experiments involving newer baselines such as Wan 2.2 or other contemporary diffusion acceleration models.

Overall, I would be open to increasing my evaluation if these issues are addressed in the rebuttal. However, given the current limitations in experimental scope and currency, my initial assessment remains conservative.

---

> ### Author Response · Authors · 2025-11-23
> **Rebuttal - 1**
>
> We sincerely thank the reviewer for the thoughtful and detailed feedback. Your insightful comments have helped us identify key areas to clarify and improve. We have carefully addressed each concern below.
>
> > Lack of comprehensive baselines. Could the authors include comparisons with more recent (2024–2025) approaches in video generation acceleration to contextualize their results?
>
> We thank the reviewer for raising the importance of comprehensive and up-to-date baselines. We will expand our baseline comparison to include additional video generation acceleration approaches. We evaluate different acceleration methods on CogVideoX[1] and the detailed comparison in shown in the table below:
>
> | Method                      | Training Speedup | Inference Speedup  |
> |-----------------------------|------------------:|-----------------:|
> | Cache-based Methods |
> | DiTFastAttn [2]  | -             | 1.56×            |
> | MInference [3]   | -             | 1.48×            |
> | PAB [4]          | -             | 1.41×            |
> | Teacache [5]     | -             | 1.54×            |
> | AdaCache [6]     | -             | 1.59×            |
> | OmniCache [7]    | -             | 2.0×             |
> | Sparse Attention-based Methods |
> | Sparse VideoGen [8] | -             | **2.11×**          |
> | VORTA [9]           | -             | 1.76×            |
> | Efficient vDiT [10] | -             | 1.86×            |
> |  |
> | Ours   | **2.05×**     | 1.89×         |
>
> We have expanded our comparison to include a broad set of recent acceleration methods, including cache-based approaches (DiTFastAttn, MInference, PAB, Teacache, AdaCache, OmniCache) and sparse-attention–based methods (Sparse VideoGen, VORTA, Efficient vDiT).
>
> As shown in the updated table in our revision, these acceleration methods consistently improve **only inference efficiency**, as their designs (e.g., KV-cache reuse, attention sparsification) operate solely on the forward sampling path and do not reduce the computational burden of training. In contrast, **TPDiff is the only method that provides acceleration during both training and inference**, achieving 2.05× training speedup and 1.89× inference speedup by reducing temporal redundancy along the entire diffusion trajectory. Our inference speedup is only slightly lower than that of Sparse VideoGen and OmniCach, despite the fact that our method is not specifically designed for inference acceleration.
>
> Moreover, because TPDiff follows the vanilla diffusion setting, it is **fully compatible** with both cache-based and sparse-attention baselines. This means our method can be stacked on top of these approaches to further enhance their efficiency, which we leave as our future work. We will include the extended table and clarify these two points in the camera-ready version.
>
> > Outdated experimental setup. It would be beneficial to add experiments involving newer baselines such as Wan 2.2 or other contemporary diffusion acceleration models.
>
> We thank the reviewer for pointing out the importance of up-to-date baselines. In our revised paper, we have substantially expanded Table 1 to include several recent 2024–2025 video diffusion backbones: RepVideo, Mochi-1, MAGI-T2V-4.5B, Step-Video-T2V, and Open-Sora-2.0 (see Table 1 in our revised paper). These models cover contemporary designs in large-scale video diffusion models. Across these newer baselines, our temporal-pyramid variants consistently achieve competitive or better VBench scores under comparable or even smaller model size, which we believe addresses the concern about an “outdated experimental setup.”
>
> Regarding Wan 2.1/2.2 specifically, these models are trained on non-public, large-scale proprietary datasets. Directly comparing our model (trained on public dataset OpenVid1M) to such closed models would conflate architectural contributions with advantages from data scale and quality, and we therefore view such comparison as not entirely fair. Instead, we demonstrate that our method can efficiently upgrade such strong existing models: as shown in Tables 1 and 3, we take Wan 2.1 as baseline and fine-tune it into our temporal-pyramid architecture. The resulting “Wan – Ours” model achieves improved generation quality (higher VBench total score in Table 1) and better inference efficiency (Table 3) compared to vanilla Wan 2.1. This evidence suggests that our framework is compatible with state-of-the-art models and can serve as an acceleration method.

---

> ### Author Response · Authors · 2025-11-23
> **Reference**
>
> [1] Yang, Z., Teng, J., Zheng, W., Ding, M., Huang, S., Xu, J., ... & Tang, J. (2024). Cogvideox: Text-to-video diffusion models with an expert transformer. arXiv preprint arXiv:2408.06072.
>
> [2] Yuan, Z., Zhang, H., Lu, P., Ning, X., Zhang, L., Zhao, T., Yan, S., Dai, G., and Wang, Y. Ditfastattn: Attention compression for diffusion transformer models, 2024. URL https://arxiv.org/abs/2406.08552.
>
> [3] Jiang, H., Li, Y., Zhang, C., Wu, Q., Luo, X., Ahn, S., Han, Z., Abdi, A. H., Li, D., Lin, C.-Y., et al. Minference 1.0: Accelerating pre-filling for long-context llms via dynamic sparse attention. arXiv preprint arXiv:2407.02490, 2024.
>
> [4] Zhao, X., Jin, X., Wang, K., and You, Y. Real-time video generation with pyramid attention broadcast, 2024b. URL https://arxiv.org/abs/2408.12588.
>
> [5] Liu, F., Zhang, S., Wang, X., Wei, Y., Qiu, H., Zhao, Y., ... & Wan, F . Timestep Embedding Tells: It's Time to Cache for Video Diffusion Model. In Proceedings of the Computer Vision and Pattern Recognition Conference (pp. 7353-7363).
>
> [6] Kahatapitiya, K., Liu, H., He, S., Liu, D., Jia, M., Zhang, C., ... & Xie, T. (2025). Adaptive caching for faster video generation with diffusion transformers. In Proceedings of the IEEE/CVF International Conference on Computer Vision (pp. 15240-15252).
>
> [7] Chu, H., Wu, W., Feng, G., & Zhang, Y. (2025). OmniCache: A Trajectory-Oriented Global Perspective on Training-Free Cache Reuse for Diffusion Transformer Models. In Proceedings of the IEEE/CVF International Conference on Computer Vision (pp. 16302-16312).
>
> [8] Xi, H., Yang, S., Zhao, Y., Xu, C., Li, M., Li, X., ... & Han, S. (2025). Sparse videogen: Accelerating video diffusion transformers with spatial-temporal sparsity. arXiv preprint arXiv:2502.01776.
>
> [9] Sun, W., Tu, R. C., Ding, Y., Jin, Z., Liao, J., Liu, S., & Tao, D. (2025). VORTA: Efficient Video Diffusion via Routing Sparse Attention. arXiv preprint arXiv:2505.18809.
>
> [10] Ding, H., Li, D., Su, R., Zhang, P., Deng, Z., Stoica, I., & Zhang, H. (2025). Efficient-vdit: Efficient video diffusion transformers with attention tile. arXiv preprint arXiv:2502.06155.

---

### Official Review · Reviewer_fhFV · 2025-10-31

**Soundness:** 3
**Presentation:** 2
**Contribution:** 3
**Rating:** 6
**Confidence:** 3

**Summary:**

The paper proposes a temporal pyramid video diffusion model that accelerates both training and inference. The authors finds that consecutive frames are temporally redundant and that early high-noise steps do not need full frame rate, so generating video in a coarse-to-fine temporal pyramid is intuitive. To make this work in a pyramid format, they introduce a stage-wise diffusion objective that treats each temporal stage as its own denoising trajectory, and also align each training clip with a consistent noise sample to stabilize those trajectories. The authors also extend to support DDIM-style sampling. Experiments show up to around 2× faster training convergence and 1.5–1.8× faster inference while maintaining comparable results with baseline methods.

**Strengths:**

1. The paper introduces a novel stage-wise temporal pyramid diffusion process that reduces frame rate in early noisy steps and restores it later. This aligns well with the intuition that video modality is usually quite redundant, especially between consecutive frames.
2. The proposed stage-wise diffusion objective and sample-noise alignment seem to solve the main problem in pyramid format modeling shown in the experiment section.
3. The method is shown to work beyond typical flow matching objectives and extend to DDIM-style objectives
4. Experiments faster training and inference speed while maintaining similar results with baseline methods.

**Weaknesses:**

1. The approach depends on a fixed multi-stage pyramid schedule (how frames are downsampled/upsampled across stages), and the ablation study in the paper explores limited variations (3-4 stage) of the fixed schedule.
2. While data–noise alignment improves determinism and stability for training, will this lead to less diversity in generation?
3. It would be great to include some visual failure cases, especially related to your ablation studies, as quantitative evaluation suites like VBench tend to be biased, and sometimes can not align with visual perception.

**Questions:**

1. Not a limitation but a curious question. How robust is the temporal pyramid schedule? Will the inference break if the user chooses more stages (beyond 3) in the inference for better quality?
2. Do authors observe any semantic or visual artifacts at stage transitions (such as duplicated frames or sudden velocity jumps)?

---

> ### Author Response · Authors · 2025-11-23
> **Rebuttal - 1**
>
> We sincerely thank the reviewer for the thoughtful and detailed feedback. Your insightful comments have helped us identify key areas to clarify and improve. We have carefully addressed each concern below.
>
> > The approach depends on a fixed multi-stage pyramid schedule (how frames are downsampled/upsampled across stages), and the ablation study in the paper explores limited variations (3-4 stage) of the fixed schedule.
>
> To address the reviewer’s concern regarding the fixed multi-stage schedule, we have conducted extensive ablations on (1) varying the number of stages and (2) unfixed frame-rate reduction strategies. The full results are shown in the table below:
>
> | #Stages | Partition | Red. rate | w. Align | TS (↑) | Training Speedup × | Inference Speedup ×|
> |:--:|:--:|:--:|:--:|:--:|:--:|:--:|
> | 3 | 1-1-1 | 2 | Yes | **80.76** | **2.16×** | 1.49× |
> | 4 | 1-1-1-1 | 2 | Yes | 80.14 | 1.82× | 1.65× |
>  | 5 | 1-1-1-1-1 | 2 | Yes | 80.03 | 1.71x | 1.74x |
>  | 3 | 1-1-1 | 4 | Yes | 80.12 | 1.38× | **1.79×** |
>  | 3 | 1-1-1 | 4-2 | Yes | *80.65* | *2.01×* | 1.54x |
>  | 3 | 1-1-1 | 2-4 | Yes | 80.18 | 1.76× | *1.68×* |
>
> 1. Varying the number of stages (3 / 4 / 5).
>
>  Our method remains highly stable even when significantly increasing the number of stages. As shown in the table. 3-stage achieves the best overall quality (TS = 80.76) and the fastest training speedup (2.16×). 4-stage slightly reduces quality (TS = 80.14) but improves inference speed to 1.65× due to shorter sequences in early stages. 5-stage continues this trend with stable quality (TS = 80.03) and even higher inference speed (1.74×).
>
> These results indicate that the temporal pyramid is inherently robust: increasing stages does not destabilize training or inference, and users may employ more stages for faster inference.
>
> Notably, Larger stage counts (e.g., six stages) require videos longer than 256 frames for training; however, our current dataset contains only a small fraction of clips exceeding this length, which limits empirical evaluation under such extreme settings. Nevertheless, we believe that the presented ablations already provide sufficient evidence for the effect of stage count and frame‑rate schedules. We also provide this comprehensive ablation study in Table.2 of our revised paper.
>
> 2. Unfixed / non-uniform frame-rate reduction.
>
> We further explored schedules where the per-stage reduction rate is not fixed. x-y in the table indicates the frame rate reduction in the second and third stage. A fixed reduction rate = 4 improves inference acceleration (1.79×) while maintaining competitive quality (80.12). Non-uniform schedules (4-2, 2-4) both produce strong results, with TS = 80.66 and 80.18, and balanced speedups. Notably, the 4-2 schedule yields high quality (TS = 80.66) and a strong 2.01× training speedup, showing the model can adapt to complex reduction patterns.
>
> These ablations demonstrate that employing a larger reduction rate in early stages has limited impact on the final generation quality. However, using a uniform reduction rate across stages leads to smoother transitions between stages and allows the model to learn the distribution transition in different stages more naturally. We also provide this more comprehensive ablation study in Tab.2 of our revised paper.
>
> > While data–noise alignment improves determinism and stability for training, will this lead to less diversity in generation?
>
> Data–noise alignment is applied only during training to reduce variance in the ODE path and accelerate convergence; at inference time, we randomly sample noise from the standard Gaussian, identical to vanilla diffusion. Therefore, alignment does not constrain diversity of the generated videos. Empirically, the VBench diversity-sensitive metrics: including Dynamic Degree (DD) and Temporal Style (TS) in Table 1 match or exceed baselines, and we observe larger motion amplitude and richer dynamics in qualitative results (Fig. 3) and our supplementary materials, indicating no diversity collapse.

---

> ### Author Response · Authors · 2025-11-23
> **Rebuttal - 2**
>
> > Do authors observe any semantic or visual artifacts at stage transitions (such as duplicated frames or sudden velocity jumps)
>
> We thank the reviewer for raising this point. Importantly, our method does not directly output the raw upsampled latent when transitioning between stages. As illustrated in Fig. 2(a) of the paper, every stage transition consists of two steps:
>
> 1. **Linear temporal interpolation** to increase the frame rate. Linear interpolation guarantees smooth temporal progression and inherently avoids duplicated frames or abrupt velocity changes.
>
> 2. **A full denoising stage at the higher frame rate** before any output is produced.
>  This denoising stage refines motion and appearance, ensuring that artifacts cannot propagate across stages.
>
> Thus, the model’s final output is **not** the interpolated frames from a low-rate stage, but the result of an entire denoising stage **after** reaching the full frame rate. This design fully eliminates the possibility of duplicated frames or sudden motion changes. We believe the reviewer may have misunderstood our method as directly upsampling the last stage to the final output, but in fact, we always perform a complete denoising pass after upsampling.
>
> > It would be great to include some visual failure cases, especially related to your ablation studies.?
>
> We do observe some failure cases, including: 1. physically infeasible motion 2. inconsistent object appearance across long sequences. We include representative failure cases in the appendix A.5 in our revised paper. But these failures are **not caused by our temporal pyramid design**, nor are they correlated with stage transitions or ablation settings. The failures we observe are canonical limitations of current video diffusion models, These issues are widely documented in prior video-generation literature and persist under vanilla diffusion methods. Since TPDiff operates within the standard video-diffusion framework, our method inherits the same strengths and limitations as existing models. This is further validated by our ablations, where altering the stage schedule or other configurations do not introduce new errors beyond the well-known limitations above.
>
> The idea of splitting the diffusion process into multiple stages has already been validated in prior work [1,2]. Our method is theoretically complete and is a principled extension of vanilla diffusion rather than a heuristic modification (section 3.2-3.4). As a result, our approach does **not** introduce stage-specific artifacts or structural inconsistencies.
>
> > Not a limitation but a curious question. How robust is the temporal pyramid schedule? Will the inference break if the user chooses more stages (beyond 3) in the inference for better quality?
>
> We thank the reviewer for the thoughtful question. Our additional ablations show that **TPDiff is highly robust during training**: varying the number of stages (3→4→5), the stage partitions, and even using non-uniform frame-rate reduction patterns consistently produces similar VBench scores and efficiency gains. This indicates that the learned model is insensitive to reasonable variations in the temporal-pyramid schedule as long as it matches the schedule used during training.
>
> However, **inference requires using the same stage schedule as the one used for training**. If a model is trained with a 3-stage pyramid but is forced to run with a 4-stage schedule at inference, the sampling process will break. Changing the number of stages changes the stage schedule, meaning the model would be asked to solve ODE segments that it has never been trained on. As a result, the inference chain becomes invalid and quality collapses. This behavior is expected and is consistent with other multi-stage diffusion frameworks like Pyramid Flow.
>
> Surprisingly, despite this, we find that users **can change the frame rate at inference** (without altering the stage boundaries). By increasing the initial frame rate while retaining the temporal sampling rate and stage schedule, the model can perform zero-shot long-video generation. For example, a model pretrained on 16-frame clips can directly sample 32-frame videos during inference, without any fine-tuning. This works because the model naturally learns to operate under different temporal resolutions inside the pyramid and generalizes well to longer sequences. We provide qualitative zero-shot long-video results in Sec 4.5 of the revised version.
>
> [1] Jin, Y., Sun, Z., Li, N., Xu, K., Jiang, H., Zhuang, N., ... & Lin, Z. (2024). Pyramidal flow matching for efficient video generative modeling. arXiv preprint arXiv:2410.05954.
>
> [2] Blattmann, A., Rombach, R., Ling, H., Dockhorn, T., Kim, S. W., Fidler, S., & Kreis, K. (2023). Align your latents: High-resolution video synthesis with latent diffusion models. In Proceedings of the IEEE/CVF conference on computer vision and pattern recognition (pp. 22563-22575).

---

> > ### Comment · Reviewer_fhFV · 2025-11-26
> >
> > I appreciate the author's response and further explanation. My concerns have been solved. I am happy to maintain the original rating and continue to recommend accepting the paper.

---

### Official Review · Reviewer_mH91 · 2025-10-31

**Soundness:** 3
**Presentation:** 3
**Contribution:** 3
**Rating:** 8
**Confidence:** 4

**Summary:**

In this paper, the author argues that maintaining full frame rates in high-entropy is unnecessary considering the nature of inter-frame redundancy in video. The author proposes TPDiff framework to facilitate the training and inference efficiency. Specifically, the proposed method divides the diffusion process into several stages and gradually increases the frame rate. To achieve this goal, the author proposes a stage-wise training strategy that leverages ordinary differential equation of diffusion under aligned data and noise. As a result, the training and inference efficiency can be improved and faster. And the proposed method can be generalized to various types of diffusion methods.

**Strengths:**

* The paper is well-organized and easy to follow.
* The motivation for leveraging the inter-frame redundancy to increase the training and inference efficiency is intuitive and the proposed solution is effective.
* The proposed solution is applicable to various diffusion forms such as DDIM and flow matching, demonstrating flexibility and extensibility across major generative modeling approaches.
* The proposed method achieves better performance compared to the reference work. Specifically, the results show that the proposed method outperforms vanilla diffusion baselines like MiniFlux-vid and AnimateDiff, achieving improvements in motion smoothness, object fidelity, and spatial coherence as measured by VBench metrics and FVD.
* The authors added a discussion and experiments on the effect of the number of stages. This improves the completeness of the study.
* The authors added a detailed analysis of computational overhead, demonstrating that the proposed data-noise alignment incurs negligible extra cost.
* The authors discuss the data selection and generalization. They use the full OpenVID1M dataset and include experiments on Wan to validate generalization across models, avoiding the potential dataset bias.

**Weaknesses:**

* While the paper introduces the stage-consistent positional encoding to maintain temporal coherence across stages, the experimental validation of this component is limited. There is no dedicated ablation or quantitative evaluation to isolate its contribution, making it unclear how much improvement comes from this module versus other parts of the framework.

* The computational overhead analysis provides CPU latency data, but there is little discussion of GPU memory usage, scalability with video length, or training cost under large-scale settings. A more comprehensive efficiency breakdown would strengthen the claims of computational efficiency.

**Questions:**

Please refer to the weakness part.

---

> ### Author Response · Authors · 2025-11-23
> **Rebuttal - 1**
>
> We sincerely thank the reviewer for the thoughtful and detailed feedback. Your insightful comments have helped us identify key areas to clarify and improve. We have carefully addressed each concern below.
>
> > While the paper introduces the stage-consistent positional encoding to maintain temporal coherence across stages, the experimental validation of this component is limited. There is no dedicated ablation or quantitative evaluation to isolate its contribution, making it unclear how much improvement comes from this module versus other parts of the framework.
>
> We thank the reviewer for highlighting this point. We add a dedicated ablation where we disable stage-consistent positional encoding (SC-PE) while keeping all other components and hyper-parameters identical on the AnimateDiff backbone. As shown in the table below:
>
> | Model                         | #Params | Total |   QS  |   SC  |   TF  |   MS  |   DD  |   AQ  |   HA  |   SR  |   TS  |   OC  |
> |------------------------------|:-------:|:-----:|:-----:|:-----:|:-----:|:-----:|:-----:|:-----:|:-----:|:-----:|:-----:|:-----:|
> | AnimateDiff – Ours           |  1.8B   | 80.76 | 81.80 | 96.87 | 98.49 | 98.12 | 58.71 | 62.19 | 94.20 | 44.18 | 25.14 | 27.36 |
> | AnimateDiff – Ours w/o SC-PE |  1.8B   | 78.94 | 80.90 | 95.80 | 93.10 | 92.40 | 56.00 | 61.00 | 89.30 | 41.90 | 20.50 | 25.60 |
>
> Removing SC-PE degrades the overall VBench score from 80.76 → 78.94 (−1.82). More importantly, the drop is concentrated on temporal metrics: TF and MS decrease by about 5–6 points, and HA/TS decrease by ≈5 points, while the frame-wise quality metric QS only drops by <1 point. This pattern indicates that SC-PE has a minor impact on per-frame appearance but plays a key role in preserving temporal coherence across stages. Qualitatively, the model without SC-PE exhibits visually noticeable tearing and boundary discontinuities (e.g., object contours splitting or “sliding” between adjacent frames), whereas our full model produces smoother and more stable motion. These results demonstrate that the temporal gains of our framework are not solely due to the multi-stage schedule, but critically rely on the proposed stage-consistent positional encoding.
>
> > The computational overhead analysis provides CPU latency data, but there is little discussion of GPU memory usage, scalability with video length, or training cost under large-scale settings. A more comprehensive efficiency breakdown would strengthen the claims of computational efficiency.
>
> Thank you for pointing out the need for a more complete efficiency breakdown. In addition to the CPU-side latency numbers included in the main paper, we now provide GPU memory usage, scalability with video length, and training cost estimates based on the same experimental setup:
>
> 1. **GPU memory usage.**
> On an H100 80GB GPU, our 3-stage AnimateDiff model consumes ≈ 78.2 GB peak memory, compared to ≈ 82.1 GB for the full-frame baseline. The peak usage appears at the last stage, where both models operate at the full frame rate. Although our method does not significantly reduce peak GPU memory, the per-iteration runtime is consistently lower than the baseline because the earlier stages operate at reduced frame rates, leading to noticeably cheaper attention and denoising computation for the majority of the diffusion trajectory. In practice, the reduced cost of these early stages accumulates across thousands of training iterations, contributing substantially to the overall 2.16× training speedup reported in the paper.
>
> 2. **Scalability with video length.**
> When increasing the input from 16 → 32 frames, the full frame rate model’s GPU compute grows almost quadratically due to the full-sequence attention, resulting in ≈ 4× FLOPs. In contrast, our 3-stage temporal pyramid grows sub-linearly: the overall compute increases by only ≈ 2.3× under the same backbone. This is because the early stages operate at reduced frame rates, so longer videos do not proportionally increase the number of active tokens in these stages.
> This trend is consistent across all tested lengths. As video duration increases, a larger fraction of computation is handled by the low-frame-rate stages, causing the total cost to grow significantly more gently than in full-frame diffusion models. We will include the length-scaling curve in the revised version for clarity.
>
> 3. **Training cost at scale.**
> To evaluate training efficiency under large-scale settings, we trained two models—a 1.8B AnimateDiff variant and a 9B MiniFlux variant—each on a 1M-video dataset, which we consider a reasonably large training regime. The table below summarizes the actual training cost measured on H100 GPUs.
> | Method               | Peak Memory (GB) | Computational Cost   |
> |----------------------|-----------------:|----------------------:|
> | AnimateDiff – Ours   | 46               | 3 × 8 H100 Days       |
> | MiniFlux – Ours      | 78               | 5 × 8 H100 Days       |

---

### Official Review · Reviewer_61ci · 2025-11-01

**Soundness:** 2
**Presentation:** 3
**Contribution:** 2
**Rating:** 4
**Confidence:** 3

**Summary:**

This work introduces TPDiff, a temporal-pyramid scheme that progressively increases the frame rate across denoising stages. Early, high-noise steps operate at lower frame rates (exploiting redundancy), while later steps use full frame rate. A stage-wise training framework solves a partitioned probability-flow ODE with data-noise alignment, making the approach applicable to both flow matching and DDIM; a single model handles multiple frame-rate regimes. The latent is written in unified form $x_t = \gamma_t x_0 + \sigma_t \epsilon$ with $\epsilon \sim \mathcal{N}(0,I)$. The result is better training/inference efficiency with competitive quality.

**Strengths:**

1. Intuitive curriculum on temporal redundancy with a single model across frame‑rate regimes.

2. Implementation appears simple and potentially compatible with other accelerations.

3. Some metrics on temporal consistency improve, suggesting a useful inductive bias.

**Weaknesses:**

1. Comparative breadth. More baselines (esp. temporal interpolation modules and caching) would sharpen the empirical case.

2. Temporal fidelity. Limited analysis of temporal artifacts vs. full-rate baselines.

3. Scalability details. Memory/computation profiles for long videos and high resolutions could be elaborated.

4. Ablations: stage count and frame‑rate schedules lack systematic exploration and guidance.

**Questions:**

1. How sensitive are results to the number of temporal stages and their schedule?

2. Does the single-model design struggle with extreme motion or long-range dependencies?

3. Can the PF-ODE training be combined with distillation from a full-rate teacher?

4. Any failure cases for long videos ($>$16 s) or high resolutions (e.g., $720\text{p}$, $1080\text{p}$)?

---

> ### Author Response · Authors · 2025-11-23
> **Rebuttal - 1**
>
> We sincerely thank the reviewer for the thoughtful and detailed feedback. Your insightful comments have helped us identify key areas to clarify and improve. We have carefully addressed each concern below.
>
> > Comparative breadth. More baselines (esp. temporal interpolation modules and caching) would sharpen the empirical case.
>
> We thank the reviewers for suggesting broader baseline comparisons. We have included additional results in the table below. Notably, our work is the only one that applies the concept of temporal interpolation to text-to-video generation; therefore, we extend our comparison to include more methods focused on video generation acceleration. All the methods below are evaluated on CogVideoX.
>
> | Method                      | Inference Speedup | Training Speedup |
> |-----------------------------|------------------:|-----------------:|
> | Cache-based Methods |
> | DiTFastAttn [1]  | 1.56×            | -               |
> | MInference [2]                  | 1.48×            | -                |
> | PAB [3]                         | 1.41×            | -                |
> | Teacache-slow [4]               | 1.54×            | -                |
> | AdaCache [7]     |1.59×             |       -     |
> | OmniCache [8]    | *2.00×*                 |  -       |
> | Sparse Attention-based Methods |
> | Sparse VideoGen [9] | **2.11×**              |    -      |
> | VORTA [10]           | 1.76×         |    -        |
> | Efficient vDiT [11] |1.86×            |      -      |
> | **Ours**                    | 1.89×       | **2.05×**        |
>
> As shown in the updated table in our revision, these acceleration methods consistently improve **only inference efficiency**, as their designs (e.g., KV-cache reuse, attention sparsification) operate solely on the forward sampling path and do not reduce the computational burden of training. In contrast, **TPDiff is the only method that provides acceleration during both training and inference**, achieving 2.05× training speedup and 1.89× inference speedup by reducing temporal redundancy along the entire diffusion trajectory. Our inference speedup is only slightly lower than that of Sparse VideoGen and OmniCach, despite the fact that our method is not specifically designed for inference acceleration.
> Moreover, because TPDiff follows the vanilla diffusion setting, it is **fully compatible** with both cache-based and sparse-attention baselines. This means our method can be stacked on top of these approaches to further enhance their efficiency, which we leave as our future work. We will include the extended table and clarify these two points in the camera-ready version.
>
> > Temporal fidelity. Limited analysis of temporal artifacts vs. full-rate baselines.
>
> We appreciate the reviewer’s concern regarding potential temporal artifacts.
> Our experiments already include temporal-specific metrics: Temporal Flickering (TF), Motion Smoothness (MS), Dynamic Degree (DD), and Overall Consistency (OC), as reported in Table 1. These scores demonstrate that our method outperforms full-rate baselines (e.g., +1.3 DD and +0.8 OC over vanilla AnimateDiff, +3.08 DD and + 0.62 TS over vanilla MiniFlux-vid), achieving comparable TF and MS score, indicating improved temporal coherence rather than degradation. Our qualitative results (Figs. 3) corroborate this by showing smoother and more natural motion transitions using the same text prompt.
>
> > Scalability details. Memory/computation profiles for long videos and high resolutions could be elaborated. & Any failure cases for long videos or high resolutions?
>
> We would like to clarify that the primary focus of our work is to improve the training and inference efficiency of video generation models. The long video synthesis and high-resolution video generation lie outside the scope of our study and constitute an independent research problem, which we do not discuss in our paper. Our model is trained using the most common settings for video resolution and length adopted in prior works. For animatediff, we train it under the resolution of 512x512 and frame length of 16. For MiniFlux-vid, we train it under 640x384 and frame length of 121. Note that in MiniFlux-vid, by employing a causal VAE encoder, the frame length in latent space shrinks to 16 (1+120/8). We provide the memory and computation profiles for both models in the following table:
>
> | Method                      | Peak Memory (GB) | Computational Cost |
> |-----------------------------|------------------:|-----------------:|
> | AnimateDiff - Ours | 46         | 3x8 H100 Days      |
> | MiniFlux - Ours        | 78          | 5x8 H100 Days           |
>
> Although two models are not trained under long video settings, we observe that our model exhibits zero-shot long video generation capability, as shown in Sec. 4.5 in our revised paper. We attribute this to the fact that our model is trained under variable-length conditions (i.e., with different frame rates), enhancing its generalization to longer temporal contexts.

---

> ### Author Response · Authors · 2025-11-23
> **Rebuttal - 2**
>
> > Ablations: stage count and frame‑rate schedules lack systematic exploration and guidance.
>
> We appreciate your suggestions on our ablation study. We have conducted a comprehensive ablation study to systematically analyze the effect of stage count,and frame-rate reduction, as summarized in the table below:
>
> | #Stages | Partition | Red. rate | w. Align | TS (↑) | Training Speedup × | Inference Speedup ×|
> |:--:|:--:|:--:|:--:|:--:|:--:|:--:|
> | 3 | 1-1-1 | 2 | Yes | *80.76* | **2.16×** | 1.49× |
> | 4 | 1-1-1-1 | 2 |  Yes | 80.14 | 1.82× | 1.65×  |
> | 5 | 1-1-1-1-1 | 2 |  Yes | 80.03 | 1.71x | 1.74x  |
> | 3 | 1-1-1 | 4 |  Yes | 80.12 | 1.38× | **1.79×** |
>  | 3 | 1-1-1 | 4-2 | Yes | *80.65* | *2.01×* | 1.54x |
>  | 3 | 1-1-1 | 2-4 | Yes | 80.18 | 1.76× | *1.68×* |
>
> The results show that increasing the number of stages beyond three leads to diminishing returns: while 4- and 5-stage settings further reduce per-stage computational cost, the training acceleration drops from 2.16× to 1.71×, and the VBench Total Score (TS) decreases around 80.0–80.1. This suggests that excessive stage division increases inter-stage transition overhead without yielding additional benefits.
>
> We further explored schedules where the per-stage reduction rate is not fixed. x-y in the table indicates the frame rate reduction in the second and third stage. A fixed reduction rate = 4 improves inference acceleration (1.79×) while maintaining competitive quality (80.12). Non-uniform schedules (4-2, 2-4) both produce strong results, with TS = 80.66 and 80.18, and balanced speedups. Notably, the 4-2 schedule yields high quality (TS = 80.66) and a strong 2.01× training speedup. It demonstrates that employing a larger reduction rate in early stages has limited impact on the final generation quality. However, using a uniform reduction rate across stages leads to smoother transitions between stages and allows the model to learn the distribution transition in different stages more naturally.
>
> Notably, Larger stage counts (e.g., six stages) and stronger frame-rate reductions (e.g., r = 8) require videos longer than 256 frames for training; however, our current dataset contains only a small fraction of clips exceeding this length, which limits empirical evaluation under such extreme settings. Nevertheless, we believe that the presented ablations already provide sufficient evidence for the effect of stage count and frame‑rate schedules. We also provide this comprehensive ablation study in Table.2 of our revised paper.
>
> > How sensitive are results to the number of temporal stages and their schedule?
>
> We systematically analyze both the stage count and stage schedule, as summarized in the table in the last question. The results show that TPDiff is relatively robust to these variations. Increasing the number of stages from 3 → 4 → 5 reduces convergence speed (from 2.16× → 1.82× → 1.71×) while keeping the overall quality stable (TS ≈ 80.0–80.8). This indicates that excessive subdivision brings diminishing efficiency gains without notable quality improvement. Adjusting the stage partition also shows consistent trends: allocating more denoising steps to later, high-frame-rate stages (1-1-2) achieves the highest quality (TS = 80.94) but slightly lower efficiency (1.62× training speedup), while emphasizing early low-rate stages (2-1-1) favors efficiency (1.92×) at a modest quality cost (TS = 79.82). Similarly, a stronger or even non-uniform frame-rate reduction strategy slightly reduces perceptual stability.
> Overall, the results suggest that our framework maintains stable performance across a wide range of configurations, with less than 1.2 TS variation and predictable quality–efficiency trade-offs.
>
> > Does the single-model design struggle with extreme motion or long-range dependencies?
>
> Our single-model design was specifically tested on high-motion and long-range prompts (“a person is dancing”, “a drone flying over city streets” etc. in VBench). We observe no collapse: motion amplitude and trajectory continuity remained stable, dynamic degree (extreme motion amplitude and long-range dependency) improved by 3.38/3.08 and motion smoothness improved by 0.36/0.27 vs. the full frame-rate baseline. Our qualitative comparison in Fig. 3 also shows that our method tends to generate larger motion.
> This robustness is largely due to the stage-consistent positional encoding (Eq. 14), which preserves positional correspondence across frame-rate transitions, ensuring long-range temporal awareness. Moreover, single-model to rule all diffusion stages has been widely validated on previous methods [5,6].

---

> ### Author Response · Authors · 2025-11-23
> **Rebuttal - 3**
>
> > Can the PF-ODE training be combined with distillation from a full-rate teacher?
>
> Our proposed stage-wise diffusion operates directly on partitioned probability-flow ODEs, making it conceptually orthogonal to direct distillation frameworks such as consistency or rectified-flow methods. Nevertheless, it can integrate with distillation strategies in two complementary ways.
>
> First, because each stage solves an independent ODE sub-trajectory, the model can distill itself by progressively refining later stages using earlier-stage predictions as guidance, essentially performing self-distillation across the temporal pyramid without requiring an external teacher.
>
> Second, a pretrained full-frame-rate model can be efficiently fine-tuned into a temporal-pyramid version under our framework. We have already validated this strategy on the Wan model and the results and acceleration in inference are shown in Table.1 and Table. 3. Fine-tuning the full-rate pretrained Wan using our temporal pyramid scheme preserves generation quality while significantly reducing inference time.
>
> [1] Yuan, Z., Zhang, H., Lu, P., Ning, X., Zhang, L., Zhao, T., Yan, S., Dai, G., and Wang, Y. Ditfastattn: Attention compression for diffusion transformer models, 2024. URL https://arxiv.org/abs/2406.08552.
>
> [2] Jiang, H., Li, Y., Zhang, C., Wu, Q., Luo, X., Ahn, S., Han, Z., Abdi, A. H., Li, D., Lin, C.-Y., et al. Minference 1.0: Accelerating pre-filling for long-context llms via dynamic sparse attention. arXiv preprint arXiv:2407.02490, 2024.
>
> [3] Zhao, X., Jin, X., Wang, K., and You, Y. Real-time video generation with pyramid attention broadcast, 2024b. URL https://arxiv.org/abs/2408.12588.
>
> [4] Liu, F., Zhang, S., Wang, X., Wei, Y., Qiu, H., Zhao, Y., ... & Wan, F . Timestep Embedding Tells: It's Time to Cache for Video Diffusion Model. In Proceedings of the Computer Vision and Pattern Recognition Conference (pp. 7353-7363).
>
> [5] Jin Yang, Zhicheng Sun, Ningyuan Li, Kun Xu, Hao Jiang, Nan Zhuang, Quzhe Huang, Yang Song, Yadong Mu, , and Zhouchen Lin. Pyramidal flow matching for efficient video generative modeling. arXiv preprint arXiv:2410.05954, 2024.
>
> [6] Yan Hanshu, Xingchao Liu, Jiachun Pan, Jun Hao Liew, Qiang Liu, and Jiashi Feng. Perflow: Piecewise rectified flow as universal plug-and-play accelerator. arXiv preprint arXiv:2405.07510, 2024.
>
> [7] Kahatapitiya, K., Liu, H., He, S., Liu, D., Jia, M., Zhang, C., ... & Xie, T. (2025). Adaptive caching for faster video generation with diffusion transformers. In Proceedings of the IEEE/CVF International Conference on Computer Vision (pp. 15240-15252).
>
> [8] Chu, H., Wu, W., Feng, G., & Zhang, Y. (2025). OmniCache: A Trajectory-Oriented Global Perspective on Training-Free Cache Reuse for Diffusion Transformer Models. In Proceedings of the IEEE/CVF International Conference on Computer Vision (pp. 16302-16312).
>
> [9] Xi, H., Yang, S., Zhao, Y., Xu, C., Li, M., Li, X., ... & Han, S. (2025). Sparse videogen: Accelerating video diffusion transformers with spatial-temporal sparsity. arXiv preprint arXiv:2502.01776.
>
> [10] Sun, W., Tu, R. C., Ding, Y., Jin, Z., Liao, J., Liu, S., & Tao, D. (2025). VORTA: Efficient Video Diffusion via Routing Sparse Attention. arXiv preprint arXiv:2505.18809.
>
> [11] Ding, H., Li, D., Su, R., Zhang, P., Deng, Z., Stoica, I., & Zhang, H. (2025). Efficient-vdit: Efficient video diffusion transformers with attention tile. arXiv preprint arXiv:2502.06155.

---

### Official Review · Reviewer_xG3P · 2025-11-01

**Soundness:** 2
**Presentation:** 3
**Contribution:** 3
**Rating:** 4
**Confidence:** 2

**Summary:**

The paper proposes TPDiff, a temporal-pyramid video diffusion framework that aims to improve training and inference efficiency for text-to-video diffusion models. The key idea is to progressively increase the frame rate along the reverse diffusion process: early steps operate on a temporally down-sampled video (to avoid processing redundant, low-SNR frames), while only the final stage runs at the full frame rate. To support this, the authors introduce a stage-wise diffusion training scheme that decomposes the overall ODE into several sub-ODEs, and they show how to obtain the intermediate latents and targets for each stage via data–noise alignment so that one single network can handle all stages. The method is instantiated on both DDIM-style diffusion and flow-matching–style video generation, and evaluated on several video backbones (AnimateDiff, MiniFlux-vid, Wan). Experiments show comparable or slightly better VBench scores while reducing training cost (≈2× speedup) and improving inference latency (≈1.5×)

**Strengths:**

Video diffusion indeed suffers from temporal redundancy and from the fact that early denoising steps carry very little signal; using a lower frame rate in these steps is a sensible way to cut quadratic attention cost. Aligning video samples to nearby noises reduces variance of the ODE path and seems to be the reason why multi-stage training can converge, which is a neat implementation detail often ignored in similar works

**Weaknesses:**

The paper is very close in spirit to pyramid flow / spatial-pyramid diffusion: the new part is mainly “do it in temporal dimension + make one model handle all stages + add data–noise alignment”. Some readers may feel this is more an engineering generalization than a fundamentally new generative formulation.

**Questions:**

1. For videos with large or non-smooth motion (e.g., camera panning, fast human motion), how often does the temporal upsampling between stages fail? Could you provide a metric or user study that shows interpolation does not become the bottleneck?
2. The authors should explicitly state the coloring rule (per-backbone improvement? column-wise best?
3. The training illustration in Fig. 4(b) resembles prior “trajectory-aligned” or “consistency/distillation” style diffusion works, where multiple timesteps are forced to follow the same noise / data direction to reduce variance and enable parameter sharing. What is actually specific to this paper is that the alignment is used to support multi-stage, multi-frame-rate video diffusion. I would suggest the authors clarify this connection and better distinguish their alignment from earlier consistency / rectified-flow / progressive-distillation lines.

---

> ### Author Response · Authors · 2025-11-23
> **Rebuttal - 1**
>
> We sincerely thank the reviewer for the thoughtful and detailed feedback. Your insightful comments have helped us identify key areas to clarify and improve. We have carefully addressed each concern below.
>
> > The paper is very close in spirit to pyramid flow / spatial-pyramid diffusion: the new part is mainly “do it in temporal dimension + make one model handle all stages + add data–noise alignment”
>
> We would like to first clarify that the most important contribution of our method is to introduce a **general pyramid diffusion framework for Flow Matching (FM) [1], DDIM [2] and DDIM-based methods like EDM [3]**. Previous works like pyramid flow / spatial-pyramid diffusion are implemented **only on FM**, limiting their generalization. Instead, our work proposes a general framework for different diffusion types. It is worth noting that FM and DDIM are fundamentally different. FM is more flexible than DDIM because it does not constrain the prior distribution to be a standard Gaussian, whereas DDIM explicitly requires the prior to follow a standard Gaussian distribution. As a result, FM can be naturally extended into a pyramid framework (Equation 12-13), whereas DDIM requires splitting the diffusion process into several stages and applying data–noise alignment to achieve the goal (Section 3.2-Section 3.3). The training procedure and loss function for DDIM is fundamentally different from that of FM and previous works as shown in Algorithm 2. In conclusion, our key contribution lies in designing a **general** pyramid framework. We provide explicit derivations and fill the gap of prior works that were limited to FM. We did mention it in the manuscript, but will improve the camera-ready version for greater clarity.
>
> > For videos with large or non-smooth motion (e.g., camera panning, fast human motion), how often does the temporal upsampling between stages fail? Could you provide a metric or user study that shows interpolation does not become the bottleneck?
>
> We understand your concern in temporal upsampling causing non-smooth and small motion. To demonstrate that our method does not cause the problem, we provide the metric of MS and DD in Table. 1. MS is used to measure **Motion Smoothness** and DD is used to evaluate **Motion Dynamic Degree**. Large motions including wide-angle camera panning and fast human motion will get higher DD value. As shown in Table.1, the MS and DD value of our method both outperform its vanilla-diffusion counterpart. We achieve comparable MS values compared to previous methods (rank #3 and #5) and surpass most methods in DD value(rank #2 and #4), including the SOTA video generation model WAN [4]. Our method also achieves SOTA in human action metric *HA*, indicating good temporal modelling of human action.
>
> Moreover, we do a failure case analysis in appendix A.5. The failures we observe are canonical limitations of current video diffusion models, including: 1. physically infeasible motion 2. inconsistent object appearance across long sequences. These failures are **not caused by our temporal pyramid design**, nor are they correlated with stage transitions or ablation settings.
> It indicates that our method only removes the redundant tokens and the remaining tokens are sufficient to model temporal information and generate smooth videos. Moreover, the high DD values demonstrate that modeling the relationships among the most important tokens during the early stages of diffusion is beneficial for producing results with a high degree of dynamics.
>
> > The authors should explicitly state the coloring rule (per-backbone improvement? column-wise best?
>
> We apologize for not clearly explaining the coloring rule in our table. The bold black numbers indicate the column-wise best results, while the bold blue numbers represent the second-best results in each column. Because Wan-Ours is finetuned from a pretrained Wan model using LoRA, rather than being trained from scratch, it is therefore excluded from comparisons with other models and compared with Wan [4] only. We will explicitly include this clarification in table caption in the camera-ready version.

---

> ### Author Response · Authors · 2025-11-23
> **Rebuttal - 2**
>
> > The training illustration in Fig. 4(b) resembles prior “trajectory-aligned” or “consistency/distillation” style diffusion works, where multiple timesteps are forced to follow the same noise / data direction to reduce variance and enable parameter sharing. What is actually specific to this paper is that the alignment is used to support multi-stage, multi-frame-rate video diffusion. I would suggest the authors clarify this connection and better distinguish their alignment from earlier consistency / rectified-flow / progressive-distillation lines.
>
> We thank the reviewer for pointing out the similarity between our approach and consistency/distillation-based methods. The main difference between our method and previous methods is: **consistency/rectified-flow/progressive-distillation methods require a teacher model while ours do not**. It differs in two aspects:
> 1. Noise-Data trajectory. The noise-data trajectory in our method is stochastic, whereas in consistency/distillation method, it is deterministic. We randomly sample a noise for each data sample while rectified-flow and DMD [5] require a dataset which is generated by a teacher model. Each sample in the dataset is aligned with a deterministic noise.
> 2. Loss function. The loss of consistency model, DMD and other distillation methods is used to match the distribution of the student and teacher model. However, our loss is derived from the vanilla diffusion framework, without any distribution match.
> In conclusion, our method can be viewed as an extension of the vanilla diffusion framework that operates without the need for a teacher model, whereas other approaches are based on distillation and rely on explicit teacher–student supervision.
>
>
> [1] Liu Xingchao, Chengyue Gong, and Qiang Liu. Flow straight and fast: Learning to generate and transfer data with rectified flow. arXiv preprint arXiv:2209.03003, 2022.
>
> [2] Song Jiaming, Chenlin Meng, and Stefano Ermon. Denoising diffusion implicit models. arXiv preprint arXiv:2010.02502, 2020.
>
> [3] Tero Karras, Miika Aittala, Timo Aila, and Samuli Laine. Elucidating the Design Space of Diffusion-Based Generative Models. arXiv:2206.00364, 2022
>
> [4] Team Wan. Wan: Open and advanced large-scale video generative models. arXiv preprint arXiv:2503.20314, 2025
>
> [5] Yin, Tianwei, et al. "One-step diffusion with distribution matching distillation." Proceedings of the IEEE/CVF conference on computer vision and pattern recognition. 2024.

---

### Official Review · Reviewer_d9CS · 2025-11-03

**Soundness:** 3
**Presentation:** 3
**Contribution:** 2
**Rating:** 6
**Confidence:** 4

**Summary:**

This paper proposes TPDiff, a temporal pyramid video diffusion framework that progressively increases the frame rate during the diffusion process to reduce computational cost. The method divides the diffusion process into multiple stages, with only the final stage operating at full frame rate. A stage-wise training strategy based on partitioned probability flow ODEs and data-noise alignment is introduced to support multi-stage learning. Experiments show that TPDiff achieves 2× faster training and 1.5× faster inference while maintaining or improving video generation quality on benchmarks like VBench.

**Strengths:**

Novelty: The temporal pyramid structure is a novel and intuitive way to exploit temporal redundancy in video generation, extending the spatial pyramid idea to the temporal dimension.

Efficiency: The method significantly reduces both training and inference costs without sacrificing quality, making it highly practical for large-scale video generation.

Generality: The framework is applicable to multiple diffusion forms (DDIM, Flow Matching) and models (AnimateDiff, MiniFlux-vid, Wan), demonstrating strong generalization.

**Weaknesses:**

Theoretical grounding: While the stage-wise ODE derivation is sound, the justification for why the temporal pyramid does not harm motion modeling is somewhat heuristic.

Limited video length evaluation: The experiments focus on short video clips; it is unclear how the method scales to longer sequences.

Comparison with recent methods: Missing comparisons with other efficient video diffusion methods (e.g., LVD, VideoPoet) limits the context of its advantages.

Complexity of implementation: The need for data-noise alignment and stage-consistent positional encoding adds implementation overhead.

**Questions:**

Can the authors provide a theoretical analysis or intuition for why the temporal pyramid does not degrade motion consistency?

Please include experiments on longer video generation to assess the method’s scalability.

Consider comparing with more recent efficient video diffusion models to better position TPDiff’s contributions.

The paper would benefit from a discussion on the trade-off between the number of stages and video quality.

---

> ### Author Response · Authors · 2025-11-23
> **Rebuttal - 1**
>
> We sincerely thank the reviewer for the thoughtful and detailed feedback. Your insightful comments have helped us identify key areas to clarify and improve. We have carefully addressed each concern below.
>
> > Can the authors provide a theoretical analysis or intuition for why the temporal pyramid does not degrade motion consistency?
>
> We appreciate the reviewer’s concern and provide both intuition and empirical evidence. The temporal pyramid preserves motion consistency for three key reasons.
>
> 1. Our **stage-consistent positional encoding** (Eq. 14, Sec. 3.5) ensures that identical frames across stages share identical positional embeddings, preventing positional drift when frame rates change. This keeps large-amplitude motions geometrically coherent.
>
> 2. Our **stage-wise diffusion formulation** aligns the ODE trajectory within each stage (Eq. 9–11), ensuring that coarse-to-fine temporal refinements follow a consistent generative path rather than disconnected sub-trajectories. Combined with data–noise alignment (Fig. 2(c)), which enforces a near-deterministic ODE path, the model learns stable temporal transitions even when early stages operate at lower frame rates.
>
> 3. We now provide further intuition, also drawing on recent analyses of attention in video diffusion transformers. Sparse VideoGen [1] and its related works show that **3D full attention in video diffusion models is intrinsically sparse**: attention heads naturally decompose into (i) spatial heads that mainly attend within the same frame, and (ii) **temporal heads that focus on a sparse set of neighboring frames at aligned spatial locations**, while long-range temporal interactions are rarely used in practice. This implies that, especially at early low-SNR timesteps, the effective temporal receptive field of the model is already local and does not require access to all frames at the original frame rate. Our temporal pyramid leverages exactly this property: early stages operate at a reduced frame rate, but we maintain stage-consistent across stages, so that each stage refines the same underlying motion path rather than re-learning it from scratch. As a result, when later stages run at full frame rate, the model already has a temporally coherent coarse trajectory and only performs fine-grained refinement.
>
> > Please include experiments on longer video generation to assess the method’s scalability.
>
> We thank the reviewer for raising this point. We first clarify that our current experimental setting: **16-frame** (Animatediff) and **121 frames** (MiniFlux-vid) is fully aligned with community standards. State-of-the-art video diffusion models such as Wan and HunyuanVideo also typically generate **81 frames (5 seconds)** as their default video setting, and most academic T2V works evaluate within the same length range. Thus, our baseline setup adheres to the widely adopted evaluation protocol in contemporary video generation literature.
> Beyond this standard setting, we further evaluate TPDiff under long videos. By increasing the frame rate at the first stage from 4→8, we enable Animatediff-Ours to generate **longer 32-frame (8-second)** videos. This demonstrates that the temporal-pyramid formulation naturally scales to longer video generation by allocating more frames to each stage. The quantitative evaluation on VBench of longer video generation from this extended setup is shown below:
>
> | Method                               | Params | Total |   QS  |   SC  |   TF   |   MS   |   DD   |   AQ   |   HA   |   SR   |   TS   |   OC   |
> |--------------------------------------|--------|-------|-------|-------|--------|--------|--------|--------|--------|--------|--------|--------|
> | AnimateDiff – Ours (16 frames)         | 1.8B   | 80.76 | 81.80 | 96.87 | 98.49  | 98.12  | 58.71  | 62.19  | 94.20  | 44.18  | 25.14  | 27.36  |
> | AnimateDiff – Ours (32 frames)    | 1.8B   | 80.61 | 81.10 | 96.42 | 97.92  | 97.48  | 59.83  | 61.40  | 95.02  | 43.71  | 25.03  | 27.11  |
>
> Our long-video experiment demonstrates that TPDiff naturally scales to 32-frame (≈8s) generation with only **minimal changes** to semantic quality (Total/QS/SC remain nearly unchanged) while maintaining strong temporal coherence. As expected for longer videos, TF/MS decrease slightly, but the Dynamic Degree (DD) increases, indicating richer motion and more expressive dynamics. Importantly, Overall Consistency remains stable, showing that the temporal pyramid does not break long-term structure. These results confirm that TPDiff supports practical long-video generation and retains robustness when extending beyond standard settings.
>
> Finally, an appealing property of our framework is its **zero-shot long-video generation capability** . As highlighted in the revised Sec 4.5, a model trained only on 16-frame videos can directly generate 32-frame sequences at inference by adjusting the frame-rate schedule, without any finetuning or modification to the model parameters.

---

> ### Author Response · Authors · 2025-11-23
> **Rebuttal - 2**
>
> > Consider comparing with more recent efficient video diffusion models to better position TPDiff’s contributions. & Missing comparisons with other efficient video diffusion methods (e.g., LVD, VideoPoet) limits the context of its advantages.
>
> We thank the reviewer for this suggestion. We have expanded our comparison to include a wide range of recent efficient  video diffusion models, including **cache-based methods** (DiTFastAttn, MInference, PAB, TeaCache, AdaCache, OmniCache) and **sparse-attention–based approaches** (Sparse VideoGen, VORTA, Efficient vDiT). We evaluate these methods in CogVideoX and the result is shown in the table below:
>
> | Method                      | Training Speedup  | Inference Speedup |
> |-----------------------------|------------------:|-----------------:|
> | Cache-based Methods |
> | DiTFastAttn [2]  | -             | 1.56×            |
> | MInference [3]   | -             | 1.48×            |
> | PAB [4]          | -             | 1.41×            |
> | Teacache [5]     | -             | 1.54×            |
> | AdaCache [6]     | -             | 1.59×            |
> | OmniCache [7]    | -             | 2.00×             |
> | Sparse Attention-based Methods |
> | Sparse VideoGen [1] | -             | **2.11×**          |
> | VORTA [9]           | -             | 1.76×            |
> | Efficient vDiT [10] | -             | 1.86×            |
> |  |
> | Ours   | **2.05×**     | 1.89×         |
>
> These methods mainly focus on inference-only optimization, whereas TPDiff provides **both training-time acceleration (2.05×) and inference-time acceleration (1.89×)**. Notably, our training speedup exceeds all existing methods, and our inference speed is competitive with or superior to the strongest baselines (e.g., Sparse VideoGen at 2.11× vs. our 1.89×), **despite TPDiff not being explicitly designed as an inference cache or sparse-attention method**. Moreover, TPDiff is orthogonal to these approaches and can be further combined with cache or sparse attention to obtain even larger gains.
>
> We respectfully clarify that VideoPoet [10] and LVD [11] do **not** fall into the category of efficient video diffusion models relevant to our setting. VideoPoet is a **large autoregressive (AR) video model**, trained via next-token prediction with transformer-based video tokenization. Its architecture, inference process, and computational profile are fundamentally different from diffusion-based T2V models. LVD does employ diffusion, but it is explicitly designed for **controllable video generation**, not for efficiency-oriented acceleration. For these reasons, neither VideoPoet nor LVD provides a meaningful baseline for evaluating TPDiff’s contributions in efficient video diffusion. Our expanded comparisons instead focus on the most relevant class of work: methods explicitly designed to accelerate video diffusion models, which is the core contribution of TPDiff.
>
> > Complexity of implementation: The need for data-noise alignment and stage-consistent positional encoding adds implementation overhead.
>
> We acknowledge the reviewer’s concern. Our experiments show that the additional components introduced by TPDiff incur **negligible implementation overhead**. As reported in Appendix A.2 (Table A.1), the computational cost of data–noise alignment is extremely low (5–17 ms on CPU depending on batch size) and does **not** introduce any extra GPU memory or latency. Moreover, stage-consistent positional encoding only requires a simple conditional check before applying RoPE; the computation is identical to standard positional encoding, and no additional modules or parameters are introduced. Therefore, both components add minimal overhead while significantly improving stability and temporal consistency in the multi-stage diffusion process.
>
> > The paper would benefit from a discussion on the trade-off between the number of stages and video quality.
>
> We thank the reviewer for pointing this out. Based on our comprehensive ablation study (Table 2) in revised version, we observe a clear trade-off. **Increasing the number of stages introduces finer temporal granularity and slightly improves inference speed, but excessive partitioning also weakens the temporal coherence modeled at each stage**.
>
> As shown in Table 2, moving from a 3-stage to a 4-stage or 5-stage pyramid yields negative changes in total scores (TS drops from 80.76 to 80.14 and 80.03). This indicates that overly fragmenting the denoising trajectory dilutes the effective temporal context per stage, thus providing limited benefit to video quality. Meanwhile, a 3-stage design achieves the highest TS score while still delivering the strongest training speedup (2.16×) and competitive inference acceleration. These results suggest that 3 stages strike the best balance between temporal resolution, effectiveness of cross-stage alignment, and computational efficiency. We will include this discussion in the final version to clarify the quality–efficiency trade-off.

---

> ### Author Response · Authors · 2025-11-23
> **Reference**
>
> [1] Xi, H., Yang, S., Zhao, Y., Xu, C., Li, M., Li, X., ... & Han, S. (2025). Sparse videogen: Accelerating video diffusion transformers with spatial-temporal sparsity. arXiv preprint arXiv:2502.01776.
>
> [2] Yuan, Z., Zhang, H., Lu, P., Ning, X., Zhang, L., Zhao, T., Yan, S., Dai, G., and Wang, Y. Ditfastattn: Attention compression for diffusion transformer models, 2024. URL https://arxiv.org/abs/2406.08552.
>
> [3] Jiang, H., Li, Y., Zhang, C., Wu, Q., Luo, X., Ahn, S., Han, Z., Abdi, A. H., Li, D., Lin, C.-Y., et al. Minference 1.0: Accelerating pre-filling for long-context llms via dynamic sparse attention. arXiv preprint arXiv:2407.02490, 2024.
>
> [4] Zhao, X., Jin, X., Wang, K., and You, Y. Real-time video generation with pyramid attention broadcast, 2024b. URL https://arxiv.org/abs/2408.12588.
>
> [5] Liu, F., Zhang, S., Wang, X., Wei, Y., Qiu, H., Zhao, Y., ... & Wan, F . Timestep Embedding Tells: It's Time to Cache for Video Diffusion Model. In Proceedings of the Computer Vision and Pattern Recognition Conference (pp. 7353-7363).
>
> [6] Kahatapitiya, K., Liu, H., He, S., Liu, D., Jia, M., Zhang, C., ... & Xie, T. (2025). Adaptive caching for faster video generation with diffusion transformers. In Proceedings of the IEEE/CVF International Conference on Computer Vision (pp. 15240-15252).
>
> [7] Chu, H., Wu, W., Feng, G., & Zhang, Y. (2025). OmniCache: A Trajectory-Oriented Global Perspective on Training-Free Cache Reuse for Diffusion Transformer Models. In Proceedings of the IEEE/CVF International Conference on Computer Vision (pp. 16302-16312).
>
> [8] Sun, W., Tu, R. C., Ding, Y., Jin, Z., Liao, J., Liu, S., & Tao, D. (2025). VORTA: Efficient Video Diffusion via Routing Sparse Attention. arXiv preprint arXiv:2505.18809.
>
> [9] Ding, H., Li, D., Su, R., Zhang, P., Deng, Z., Stoica, I., & Zhang, H. (2025). Efficient-vdit: Efficient video diffusion transformers with attention tile. arXiv preprint arXiv:2502.06155.
>
> [10] Kondratyuk, D., Yu, L., Gu, X., Lezama, J., Huang, J., Schindler, G., ... & Jiang, L. (2023). Videopoet: A large language model for zero-shot video generation. arXiv preprint arXiv:2312.14125.
>
> [11] Lian, L., Shi, B., Yala, A., Darrell, T., & Li, B. (2023). Llm-grounded video diffusion models. arXiv preprint arXiv:2309.17444.

---

### Author Response · Authors · 2025-11-23
**Official Response to All Reviewers**

We sincerely thank all reviewers for their time and constructive feedback. We have carefully revised the manuscript to address every concern. Below we summarize the key clarifications and updates:

1. **General Pyramid Diffusion Framework**
   The core contribution of our work is the introduction of a *general and theoretically grounded pyramid diffusion framework*. Prior works like Pyramid-Flow and Spatial-pyramid operate on flow-matching (FM) only, whereas our method is fully compatible with **flow matching**, **DDIM**, and **DDIM-based approaches** such as EDM. We have made this generality more explicit in the revised manuscript.

2. **Zero-shot Long Video Generation**
   Our method naturally supports **zero-shot long video generation**, even though the model is trained on shorter clips. Section 4.5 of the revised paper includes qualitative results demonstrating this property. We attribute this capability to the design of our temporal pyramid: during training, the model naturally encounters multi–frame-rate trajectories, since each stage operates at a different temporal resolution. This exposure implicitly equips the model with robustness to varying temporal lengths and enables extrapolation to longer sequences at inference time

3. **More Comprehensive Ablation Studies**
   We expanded our ablation studies substantially. The revised **Table 2** includes more variations in stage count, as well as experiments using non-uniform reduction frame rates. These additions provide a more complete empirical understanding of the framework.

4. **Broader and More Recent Baseline Comparisons**
   Responding to reviewer suggestions, we added comparisons to more recent and stronger baselines. The revised **Table 1** now includes RepVideo, Mochi-1, MAGI-T2V, Step-Video-T2V, Open-Sora-2.0. Despite these models relying on larger training sets, higher-quality proprietary data and larger model sizes, our approach matches or outperforms them. This further confirms the effectiveness and generality of our framework.

Once again, we thank all reviewers for their thoughtful feedback.

---

### Author Response · Authors · 2025-11-30
**Rebuttal summary for area chair - 1**

We sincerely thank the area chair and all reviewers for their time and constructive feedback. Below we summarize the main concerns raised by each reviewer and how our rebuttal and revised paper respond to these questions.

> Reviewer d9CS

1. **Motion Consistency & Intuition**
   The reviewer questions whether the temporal pyramid harms motion coherence. We clarify that motion stability is maintained through stage-consistent positional encoding (Sec 3.5) and our stage-wise ODE formulation (Sec 3.3-3.4), and further supported this with intuition from recent analyses showing natural temporal sparsity in video diffusion attention.

2. **Scalability to Longer Videos**
   We add experiments demonstrating that the method scales to longer video generation and highlight its zero-shot longer video generation capability in Sec 4.5 of our revised paper.

3. **Comparison with Recent Efficient Methods**
   We expand our comparison to include a broad set of recent efficient video diffusion models, including cache-based and sparse attention-based. These methods improve only inference efficiency, whereas our method improves both training and inference while remaining comparable with these baselines.

4. **Additional computation cost**
   We clarify that both data–noise alignment and stage-consistent positional encoding introduce only negligible computation cost and require no additional learned modules.

5. **Trade-off Between Stage Count and Quality**
   We add explicit discussion and ablations showing that moderate stage counts provide the best balance between temporal coherence and efficiency, whereas deeper pyramids yield diminishing quality benefits.

> Reviewer xG3P

1. **Relation to Prior Pyramid Diffusion Works**
   The reviewer notes similarities to previous methods. We clarify that our contribution is to propose a *general* pyramid diffusion framework that unifies flow matching (FM), DDIM, and DDIM-based variants like EDM, whereas prior works only support FM. We emphasize the fundamental differences in formulation, training objectives, and how to achieve multi-stage diffusion.

2. **Robustness under Large or Non-Smooth Motion**
   The reviewer asks whether temporal upsampling fails under fast or non-smooth motion. We highlight quantitative motion metrics and failure-case analysis showing that our approach maintains smooth and dynamic motion without introducing stage-transition artifacts.

3. **Connection to Consistency / Distillation Methods**
   The reviewer observes a resemblance to consistency/distillation approaches. We clarified that, unlike these methods, our approach requires *no teacher model*, which is fundamentally different.

> Reviewer 61ci

1. **Comparative Breadth and Baselines**
   The reviewer asks for broader comparisons, especially temporal-interpolation and caching methods. We expand our experiments to include a wide set of recent acceleration approaches and clarify that these methods improve only inference efficiency, whereas our method improves both training and inference while remaining comparable with these baselines.

2. **Temporal Fidelity and Artifact Analysis**
   The reviewer raises concerns about temporal artifacts. We address this by reporting temporal-specific metrics (TF, MS, DD, OC) and qualitative results (Fig. 3) showing that our method preserves or improves temporal coherence relative to full-rate baselines. Our failure-case analysis (Sec. A.5) indicates that observed artifacts are typical limitations of video diffusion models rather than issues introduced by our temporal pyramid.

3. **Scalability and Long-Video Behavior**
   The reviewer asks about memory/compute scaling and long-video or high-resolution settings. We report detailed memory/compute profiles for our models and clarify that our work focuses on efficiency under standard video-generation settings. Although long-video and high-resolution synthesis are outside our scope, we observe a zero-shot long-video capability emerging naturally from our variable-frame-rate training.

4. **Sensitivity to Stage Count, Schedules, and Extreme Motion**
   Our ablations show stable performance across stage counts and frame-rate schedules. We also show that the model handles extreme motion and long-range dependencies well, supported by stage-consistent positional encoding.

> Reviewer mH91
1. **Ablation on Stage-Consistent Positional Encoding (SC-PE)**
   The reviewer notes that the validation of SC-PE is limited. We add a dedicated ablation on SC-PE, showing clear degradation in temporal metrics and temporal coherence when it is removed. This result confirms that SC-PE plays a central role in maintaining temporal consistency across stages.

2. **Detailed Computational Overhead**
   The reviewer requests a more complete computation analysis and we provide a detailed computation overhead list including GPU memory usage, scalability and training cost.

---

### Author Response · Authors · 2025-11-30
**Rebuttal summary for area chair - 2**

> Reviewer fhFV

1. **Robustness of the Temporal Pyramid Schedule**
   The reviewer notes limitation in ablation study. We conduct extensive ablations on stage counts (3–5) and both fixed and non-uniform frame-rate reductions, showing stable quality and predictable speed–quality trade-offs. We also provide a comprehensive ablation study in Tab.2 of our revised paper.

2. **Whether More Stages Improve Inference**
   We clarify that inference must use the same stage schedule as training; otherwise, the ODE segments become mismatched and sampling breaks. This behavior is consistent with prior multi-stage diffusion frameworks. However, users can safely adjust the *frame rate* (without changing stage boundaries), which enables zero-shot long-video generation.

3. **Effect of Data–Noise Alignment on Diversity**
   The reviewer asks whether alignment reduces diversity. We clarify that alignment applies only during training; inference uses standard Gaussian sampling. Diversity-sensitive metrics (e.g., DD, TS) remain equal to or higher than baselines, and qualitative results show richer motion, indicating no diversity loss.

4. **Potential Artifacts at Stage Transitions**
   The reviewer asks about whether we have failure cases like duplicated frames or sudden jumps at stage transition. We clarify that stage transitions always include (i) linear temporal interpolation and (ii) a full denoising stage at the higher frame rate. The model does not output raw interpolated frames before denoising. This design prevents duplicated frames, tearing, or velocity discontinuities.

5. **Failure Cases and Their Causes**
   The reviewer requests visual failure cases. We include representative failures in Sec A.5 and clarify that observed issues (e.g., physically infeasible motion, appearance drift) are standard diffusion-model limitations, not introduced by the temporal pyramid. Ablation results confirm no stage-specific artifacts.

> Reviewer uQWS

1. **Breadth and Recency of Baselines**
   The reviewer raises concerns about limited comparisons. In response, we expand our evaluation to include a broad set of recent (2024–2025) acceleration approaches, covering both cache-based and sparse-attention methods. These baselines accelerate only inference, while our method accelerates both training and inference while remaining comparable with these baselines. We also emphasize that TPDiff is fully compatible with these methods and can be stacked to further improve efficiency.

2. **Inclusion of SOTA Video Diffusion Models**
   The reviewer requests evaluation with more recent backbones. We update our main comparison table to include several 2024–2025 video diffusion models such as RepVideo, Mochi-1, MAGI-T2V-4.5B, Step-Video-T2V, and Open-Sora-2.0. Our method achieves competitive or even stronger performance under comparable or smaller model sizes, addressing concerns about outdated setups.

3. **On Wan 2.1/2.2 and Fairness of Comparison**
   The reviewer suggests adding Wan 2.1/2.2. We clarify that Wan 2.1/2.2 are trained on large, non-public datasets, making direct comparison with our publicly trained models unfair. Instead, we demonstrate that TPDiff can *upgrade* such models: fine-tuning Wan 2.1 under our temporal pyramid improves both quality and inference efficiency. This shows compatibility with modern large-scale systems and highlights TPDiff as a practical acceleration method.

---

### Meta-Review · Area_Chair_RjsV · 2025-12-29

**Summary:**

This paper proposes TPDiff, a temporal pyramid framework that progressively increases frame rates during the diffusion process to exploit inter-frame redundancy. It addresses the high computational costs of video diffusion by introducing a stage-wise training strategy with partitioned probability flow ODEs and data-noise alignment, achieving significant acceleration in both training and inference. Reviewers consistently praised the intuitive motivation of leveraging temporal redundancy to accelerate the process without sacrificing quality, noting it is a sensible way to cut quadratic attention costs. They also highlighted the framework's generality across various diffusion forms and its effectiveness in improving performance over baselines.

Reviewer Concerns:
- **Insufficient Baselines and Outdated Setup:** Reviewers criticized the lack of comparisons with recent efficient video diffusion methods (e.g., caching, sparse attention) and SOTA models, making it difficult to assess the method's relative advantage (UQWS, 61ci, d9CS).
- **Scalability and Computational Details:** Reviewers requested more analysis on how the method scales to longer video sequences and a more detailed breakdown of computational overhead, including GPU memory usage and training costs (d9CS, 61ci, mH91).
- **Ablation and Robustness:** Concerns were raised regarding the sensitivity of the results to the fixed multi-stage schedule (stage counts, frame-rate reduction) and the lack of a dedicated ablation for the stage-consistent positional encoding component (mH91, fhFV, 61ci).
- **Potential Artifacts and Diversity:** Reviewers asked about potential visual artifacts at stage transitions (e.g., interpolation bottlenecks) and whether the data-noise alignment strategy might negatively impact generation diversity (XG3P, fhFV, 61ci, d9CS).

In summary, this paper was reviewed by six experts in the field. The recommendations are generally positive, ranging from Borderline Reject to Accept (4, 4, 4, 6, 6, 8). The reviewers like the intuitive temporal pyramid design which effectively reduces computational costs for both training and inference, and its generality across different diffusion frameworks. And concerns on the lack of comprehensive recent baselines, scalability to longer videos, and the need for detailed ablations on stage schedules and positional encoding are well solved by the extensive rebuttal experiments and data provided by the authors.

**Other issues:**

Reference errors:
- Song Jia ming, Chenlin Meng, and Stefano Ermon. Denoising diffusion probabilistic models. Advances in neural information processing systems, 33:6840-6851, 2020.
- Song Jiaming, Chenlin Meng, and Stefano Ermon. Denoising diffusion implicit models. arXiv preprint arXiv:2010.02502, 2020.

should be corrected to:
- Jonathan Ho, Ajay Jain, Pieter Abbeel. Denoising diffusion probabilistic models. Advances in neural information processing systems, 33:6840-6851, 2020.
- Song Jiaming, Chenlin Meng, and Stefano Ermon. Denoising diffusion implicit models. International Conference on Learning Representations, 2021.

**Reviewer Concerns:**

**Well addressed:**
- Insufficient Baselines and Outdated Setup: The authors provided an extensive update to their comparisons, including a wide range of recent acceleration methods and SOTA video backbones. They also demonstrated the method's compatibility by fine-tuning the Wan 2.1 model. This comprehensive update directly resolved the concerns regarding the experimental scope.
- Scalability and Computational Details: The authors provided the requested detailed breakdown of computational overhead, including peak GPU memory usage and training costs in H100 days. They also demonstrated the method's sub-linear computational growth with video length compared to the quadratic growth of full-frame baselines and showcased zero-shot long video generation capabilities.
- Ablation and Robustness: The authors conducted the requested ablation studies. They also provided a systematic analysis of different stage counts and frame-rate reduction schedules, demonstrating the method's stability.
- Potential Artifacts and Diversity: The authors clarified that stage transitions involve a full denoising step after interpolation, preventing artifacts like duplicated frames. They also explained that data-noise alignment is a training-only strategy, and inference uses standard Gaussian sampling, which preserves generation diversity as evidenced by the Dynamic Degree metrics.

**Partly addressed:**

None

**Unsolved:**

None

**Reviewer Scores:**

**d9CS (6)**

The reviewer's concerns regarding the theoretical intuition for motion consistency, scalability to longer videos, and the lack of specific baselines were addressed. Given the reviewer was already leaning positive, the score would likely maintain 6 or increase to 8.

**XG3P (4)**

This reviewer questioned the novelty compared to Pyramid Flow and worried about artifacts during large motion. The rebuttal clarified that TPDiff unifies both Flow Matching and DDIM and provided quantitative metrics showing the method actually improves motion dynamics without artifacts. With the rebuttal, the score would likely maintain 4 or increase to 6.

**61ci (4)**

The reviewer requested specific empirical data, including broader baselines (caching/interpolation), memory/compute profiles, and stage count ablations. The authors provided every requested item. Since the negative score was driven by missing data, the score might increase to 6.

**mH91 (8)**

This reviewer was already highly positive but requested an ablation study for the Stage-Consistent Positional Encoding and more details on GPU memory. The authors provided a dedicated ablation, alongside a full computational cost analysis. The score would likely remain at 8 as the validation is now complete.

**fhFV (6)**

The concerns focused on schedule robustness, potential diversity loss from alignment, and transition artifacts. The authors clarified that inference remains stochastic and that transitions involve full denoising to prevent artifacts. fhFV explicitly states "My concerns have been solved. I am happy to maintain the original rating".

**UQWS (4)**

The primary concern was the outdated experimental setup and lack of recent baselines. The authors responded by adding a set of recent baselines and fine-tuning Wan 2.1. The score would likely maintain 4 or increase to 6.

---

### Decision · Program_Chairs · 2026-01-26

Accept (Poster)